# Deglacial increase of seasonal temperature variability in the tropical ocean

Lars Wörmer[1✉], Jenny Wendt[1], Brenna Boehman[1,4], Gerald H. Haug[2,3] & Kai-Uwe Hinrichs[1]

The relatively stable Holocene climate was preceded by a pronounced event of abrupt warming in the Northern Hemisphere, the termination of the Younger Dryas (YD) cold period[1,2]. Although this transition has been intensively studied, its imprint on low-latitude ocean temperature is still controversial and its effects on sub-annual to decadal climate variability remain poorly understood[1,3,4]. Sea surface temperature (SST) variability at these timescales in the tropical Atlantic is expected to intensify under current and future global warming and has considerable consequences for environmental conditions in Africa and South America, and for tropical Pacific climate[5–8]. Here we present a 100-μm-resolution record obtained by mass spectrometry imaging (MSI) of long-chain alkenones in sediments from the Cariaco Basin[9–11] and find that annually averaged SST remained stable during the transition into the Holocene. However, seasonality increased more than twofold and approached modern values of 1.6 °C, probably driven by the position and/or annual range of the Intertropical Convergence Zone (ITCZ). We further observe that interannual variability intensified during the early Holocene. Our results demonstrate that sub-decadal-scale SST variability in the tropical Atlantic is sensitive to abrupt changes in climate background, such as those witnessed during the most recent glacial to interglacial transition.

The warm and relatively stable climate of the Holocene has facilitated the development of modern ecosystems and the proliferation of human societies and their cultural diversification. Its onset, however, was associated with an event of abrupt climate change. The boundary between the Pleistocene and the Holocene is defined by the sudden Northern Hemisphere warming that terminated the YD cold spell[12]. The YD lasted from about 12.9 to 11.7 thousand years before 2000 AD (kyr b2k)[13] and was triggered by a reduction of the Atlantic meridional overturning circulation resulting from freshwater discharge at higher latitudes[14]. Its effects quickly propagated, globally affecting hydroclimate and temperature[1,2]. Although correlative cooling was predominant across the Northern Hemisphere, the Southern Hemisphere, especially in the high latitudes, witnessed warming in what has been defined as the bipolar seesaw[15]. The annually laminated (varved) sediments from the Cariaco Basin, an anoxic oceanic basin located off Venezuela, have been crucial in identifying the tropical response to the YD–Holocene transition. A dry YD was succeeded by a wetter early Holocene[16], resulting in vegetation change[17]. The end of the YD also witnessed changes in primary productivity[11,18,19] and phytoplankton community composition[20]. These phenomena are explained by a northward migration of the ITCZ, which resulted in increased precipitation, but reduced trade winds and upwelling in the area[11,17,19].

With respect to SST, the reconstructed pattern of change in the lower latitudes is more heterogeneous, and a greater impact of the YD on the hydrological cycle than on SST is assumed[1]. In the western tropical North Atlantic (TNA), different reconstructions have yielded inconsistent results, with evidence of both a slightly warmer YD from molecular proxies and planktonic foraminifera, consistent with a decrease of northward heat transport[4,21], but also a comparatively cool YD, as recorded by foraminifera in the Cariaco Basin[3].

These SST reconstructions record changes in mean states, averaging decades or centuries into single data points. The forcing of seasonal to interannual SST variability during this and other notable climatic transitions, however, remains unexplored. Perturbations on these timescales are, nevertheless, highly relevant: SST variability in the TNA profoundly affects precipitation in Africa and South America, including catastrophic droughts in Northeast Brazil[8], modulates the incidence of hurricanes[7], and influences tropical Pacific climate, particularly El Niño–Southern Oscillation (ENSO)[5]. Understanding TNA SST variability across changing climate backgrounds is thus of relevance, especially considering that an intensification under greenhouse warming has been anticipated[6].

We analysed the well-established $U_{37}^{K'}$ SST proxy, based on the distribution of haptophyte-derived alkenones[22], by means of MSI at 100-μm resolution in a 60-cm section of the well-dated core MD03-2621 from the Cariaco Basin. This section spans an age of approximately 11.9 to 11.2 kyr b2k and thus includes the YD–Holocene transition[11]. The resulting SST record provides insights into seasonal to interannual SST variability during the most recent glacial to interglacial transition.

## Average annual SST

Despite the extensive environmental changes related to the northward shift of the ITCZ, which caused the abrupt change in sediment

[1]MARUM – Center for Marine Environmental Sciences and Faculty of Geosciences, University of Bremen, Bremen, Germany. [2]Max Planck Institute for Chemistry, Mainz, Germany. [3]Department of Earth Sciences, ETH Zürich, Zürich, Switzerland. [4]Present address: MIT-WHOI Joint Program in Oceanography/Applied Ocean Science & Engineering, Cambridge and Woods Hole, MA, USA. ✉e-mail: lwoermer@marum.de

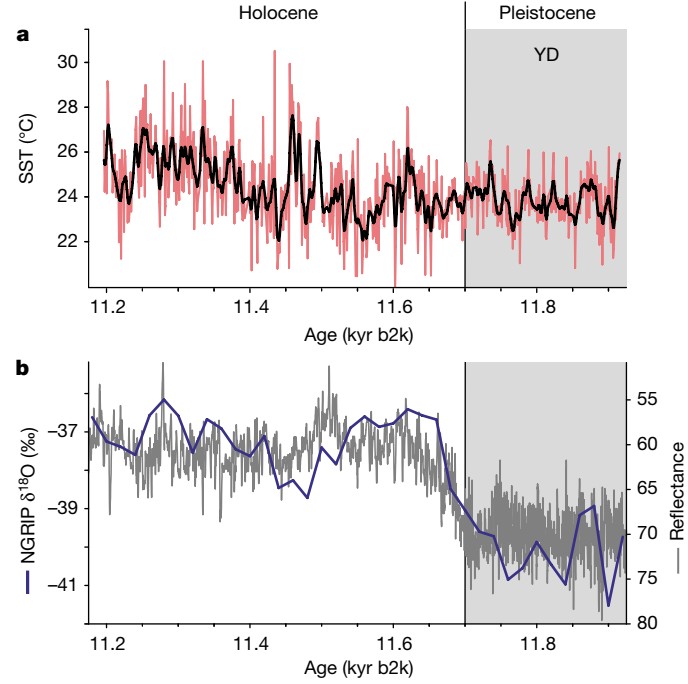

**Fig. 1 | Reconstructed SST in the tropical Cariaco Basin across the YD–Holocene transition based on the $U^{K'}_{37}$ proxy analysed through MSI.** **a**, Annually averaged SST (red line) and 15-point Gaussian smoothing (black line). **b**, Changing sediment reflectance[11] is an indicator of the environmental change in the Cariaco Basin associated with the transition from the cold YD to a milder Holocene, as recorded in $\delta^{18}O$ values in Greenland ice cores[13]. The beginning of the Holocene (11.7 kyr b2k) is indicated as defined by Rasmussen et al.[13].

reflectance (Fig. 1b) and varve thickness[11,19], our high-resolution reconstructions indicate that annually averaged SST remained constant across the YD–Holocene transition (Fig. 1a). In the 200 years before and after the reflectance-based midpoint of the YD–Holocene transition (11.673 kyr b2k)[11], the average SST is 23.8 ± 1.0 °C and 23.8 ± 1.6 °C, respectively. This is consistent with conventional $U^{K'}_{37}$ analyses performed in the present study and with those previously reported by Herbert and Schuffert[23] but different from the YD cooling described by Lea et al.[3] (Extended Data Fig. 1). We argue that the transition into the Holocene did not have an imprint on the average annual SST and that conflicting, low-resolution SST records from the Cariaco Basin and the TNA can be explained by seasonal effects and changes to water-column stratification. Such seasonal effects are explored in a dedicated section below.

Three prominent SST maxima are observed between about 11.50 to 11.45 kyr b2k and thus coincide in time with the '11.4 ka cold event', or Preboreal oscillation (PBO) (Extended Data Fig. 2). The PBO was caused by a weakening of the thermohaline circulation[24,25], and is registered in records from Europe and North America as a shift towards dryer, colder conditions[25,26]. These maxima are discussed in more detail in Methods, under the section titled 'Decadal-scale to centennial-scale SST changes during the YD–Holocene transition and in the early Holocene'.

## Interannual SST variability

Although apparently the longer-term SST trend was barely affected during the YD–Holocene transition, short-term variability increased. Frequency analysis of the annually resolved $U^{K'}_{37}$ SST record shows persistent centennial (120-year) and multidecadal (42-year) cycles. The variability at sub-decadal frequencies is weak during the late YD but substantially strengthens at 11.66 kyr b2k ($P$ = 0.006) and remains prominent in the Holocene section (Fig. 2a–c and Extended Data Fig. 3).

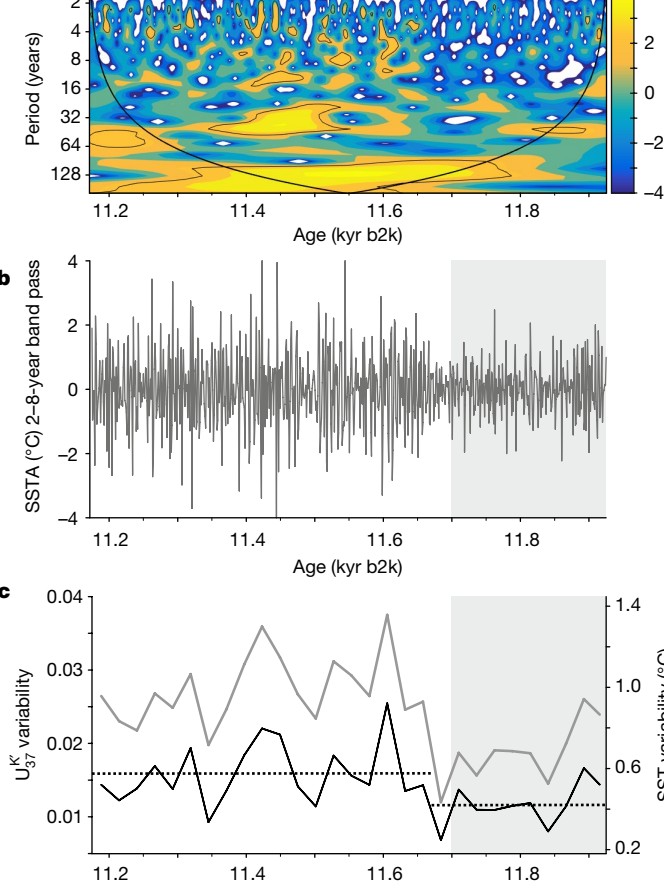

**Fig. 2 | Interannual SST variability during the YD–Holocene transition.** **a**, Continuous Morlet wavelet power of the $U^{K'}_{37}$ SST series. Contour lines denote the 95% significance level against red noise and the line marks the cone of influence. **b**, SST anomaly (SSTA) band-pass-filtered for a period of 2–8 years. **c**, SST variability in the 2–8-year window for 25-year intervals. The grey line represents raw data, whereas the black line shows data corrected for analytical variability (see Methods, under the sections titled 'SST reconstruction with yearly resolution' and 'The effect of changing sedimentation rate on reconstructed interannual SST variability during the YD–Holocene transition'). The dotted lines represent average variability before/after 11.66 kyr b2k.

The modern tropical Atlantic presents strong interannual anomalies, mostly expressed in a zonal pattern analogous to the Pacific El Niño and a meridional pattern that results from the mutual interaction of the latitudinal SST gradient, the ITCZ position and the strength of trade winds[27,28].

This variability is driven by local processes, but can also be forced by ENSO and the North Atlantic Oscillation[29,30]. ENSO has a strong teleconnection to TNA SST: El Niño events in boreal fall and winter tend to result in positive SST anomalies during the following spring in the TNA[31–33] (Extended Data Fig. 4). The muted interannual SST variability during the late YD thus seems to stand in conflict with the hypothesis that the meltwater-induced collapse of overturning circulation during the YD would have resulted in increased ENSO amplitude[34,35] and with estimations of a strengthened mid-YD ENSO based on individual foraminifera analysis of discrete samples with ages of 12.5 and 12.2 kyr b2k (refs. [36,37]). However, our data reflect approximately the last two centuries of the YD, when the meltwater effect might have already ceased. Cheng et al.[2] recently claimed that the YD termination might have started in the Southern Hemisphere (at around 11.9 kyr) or the tropical Pacific (at around 12.3 kyr), owing to a shift from El Niño to La Niña-like conditions.

This shift would have induced a gradual strengthening of the Atlantic meridional overturning circulation until reaching a tipping point that led to the abrupt rise in North Atlantic temperature. A muted ENSO teleconnection during the late YD would thus be feasible and might have contributed, among other factors, to reduced SST variability in the TNA. Subsequent strengthening of SST variability in the early Holocene might be related to the northward shift of the ITCZ or to sea ice retreat.

## Seasonal SST variability

In our record, sub-annual SST variability can be assessed by deconvoluting the SST signal into upwelling and non-upwelling seasons. This was achieved by combining information on sediment colour, elemental composition and $U^{K'}_{37}$ values in each μm-sized spot (Methods, sections titled 'Assessment of SST seasonality' and 'Varve formation and alkenone deposition in the sediments of the Cariaco Basin during the YD–Holocene transition'). Using micro-X-ray fluorescence spectroscopy analysis of sampled spots congruent to those analysed by MSI, we confirmed that the investigated laminae couplets represent annual cycles, as already proposed by Hughen et al.[18]: darker layers are enriched in Fe, Ti and Ca and correspond to the rainy, non-upwelling (summer/fall) season depositing terrigenous material and biogenic CaCO$_3$ from foraminifera or coccolithophores. Si abundance is highest in lighter layers and corresponds to the increased biogenic opal production by diatoms during the upwelling (winter/spring) season (Fig. 3b and Extended Data Fig. 5). This blueprint of seasonality was used to assess changes in alkenone abundance (Extended Data Fig. 6) and $U^{K'}_{37}$-based SST reconstruction. Light layers record lower SST values, consistent with upwelling-induced cooling, whereas darker layers show higher values (Fig. 3c and Extended Data Fig. 7). Deconvolution of reconstructed SST, based on sediment colour, enabled us to calculate the seasonality of SST, defined as the difference between average SST in the non-upwelling and upwelling seasons. SST seasonality substantially strengthened into the Holocene ($P < 0.001$) and this increase is robust across a range of sediment colour values chosen to separate data into upwelling and non-upwelling seasons (Extended Data Fig. 8). Fitting changes in seasonality to a ramp (see Methods, section titled 'Assessment of SST seasonality') results in an abrupt increase from 0.8 to 1.8 °C at 11.64 kyr b2k, whereas imposing a more gradual increase results in a 160-year trend from 0.6 to 1.8 °C (Fig. 3a). Reconstructed early Holocene seasonality is thus similar to the modern Cariaco Basin (1.6 °C).

This increase in seasonality is concurrent with the change in sediment reflectance and we posit that both were forced by the position of the ITCZ. Annual SST variability in the tropical Atlantic is driven by feedbacks between ocean and atmosphere[38–40], and SST maxima are coupled to the seasonal migration of the ITCZ. Its northernmost displacement in summer/fall results in warmest surface waters in the Caribbean Sea. Decreased seasonality during the late YD can thus be explained by a southward shift of the mean position of the ITCZ and/or by a contraction of its annual range.

This effect was probably further reinforced by local features in the Cariaco Basin. The basin sits within the current area of migration of the Atlantic ITCZ, more precisely at its northernmost limit. This translates into a strong seasonal cycle: in summer and fall, heavy precipitation is related to the ITCZ being positioned over the catchment area of the basin and results in intense discharge by local rivers[41]. In winter and spring, as the ITCZ migrates southward, it allows for strong trade winds, increased upwelling, weakened stratification and highest primary productivity and export of biogenic material[42]. We argue that, as the ITCZ moved northward during the transition into the Holocene and/or expanded its annual range, the summer/fall season received larger freshwater input and became less influenced by regional windiness and upwelling[11,16,19]. This allowed the development of density stratification, with warm surface layers, as opposed to the colder mixed water column of the upwelling season. The incipient strengthening

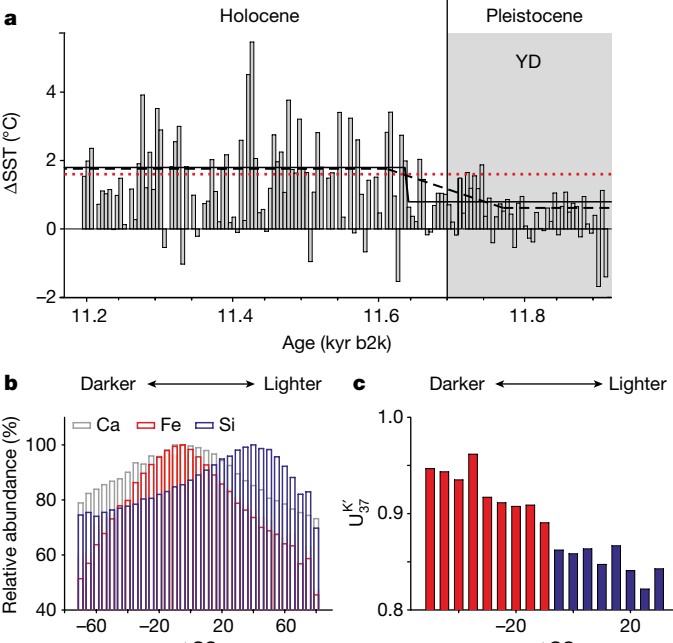

**Fig. 3 | Reconstructed SST seasonality across the YD–Holocene transition based on the $U^{K'}_{37}$ proxy analysed through MSI. a**, SST seasonality calculated as the difference between $U^{K'}_{37}$ SST attributed to the non-upwelling and upwelling seasons in 5-year intervals. Increase in Holocene seasonality is fitted to an abrupt and a more gradual ramp (solid and dashed black lines). The dotted red line represents modern Cariaco SST seasonality (1.6 °C). **b,c**, Seasonality was evaluated by assigning molecular proxy data from each spot to a season of deposition based on the sediment colour of the spot. Elemental (**b**) and $U^{K'}_{37}$ (**c**) data from an exemplary 5-cm slice (490–495 cm below seafloor, 11.39–11.50 kyr b2k) binned according to sediment colour (greyscale, GS). For better visualization, each bin encompasses five GS units and includes at least 25 successful $U^{K'}_{37}$ analyses. The red and blue bars in **c** are attributed to non-upwelling and upwelling seasons, respectively. GS is shown as ΔGS, that is, the difference from the median GS of the slice.

of SST seasonality would have been further supported by a maximum in insolation seasonality[43] (Extended Data Fig. 9a).

In the modern Cariaco Basin, temperature has been identified as a chief driver of phytoplankton composition[44], for example, warmer temperatures exert a negative effect on most diatoms. Stronger SST seasonality and a warmer non-upwelling season thus can be related to a more pronounced annual succession in the phytoplankton composition and to the shift from a diatom-dominated YD to a coccolithophore-dominated Holocene[20,45]. We further suggest that these changes in seasonality will have affected previous, lower-resolution SST reconstructions. The abrupt warming inferred by Lea et al.[3] might actually reflect increased SST in the thermally stratified water column of the non-upwelling season (Methods, section titled 'Effect of changing seasonality on YD and early Holocene SST records from the western TNA' and Extended Data Fig. 9b).

Bova et al.[46] have proposed that climatic events such as the Holocene and last interglacial thermal maxima are actually associated with large seasonal effects but weak annual SST changes. Our dataset provides proxy-based evidence of such seasonal effects in the TNA during the last abrupt transition to a warmer climate at the Pleistocene–Holocene boundary. Our record further describes the strengthening of TNA interannual SST variability over this boundary. By revealing these previously hidden sources of SST variability, we conclude that the high-frequency component of the climate system can be especially sensitive to changes in climate background. Assessing this variability over critical climate transitions through reconstructions has been hampered in the past by

insufficient resolution in proxy records, but is now feasible through MSI-based analysis of molecular proxies and its combination with other high-resolution techniques.

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

## Methods

### Study site

The Cariaco Basin, located on the continental shelf off Venezuela, is a large (about 160 km long and about 65 km wide) depression, composed of two approximately 1,400-m-deep sub-basins. It is partially isolated from the Caribbean Sea by a series of sills with depths of less than 150 m (ref. [47]). This limits renewal of deep water in the basin and, paired with the high oxygen demand resulting from intense surface primary productivity, leads to anoxic waters below a depth of about 275 m at present[47,48].

The marked seasonality in the Cariaco Basin, combined with anoxic bottom waters that effectively prevent bioturbation, results in the accumulation of annually laminated (varved) sediments. As sediments are varved for the last deglaciation and the Holocene, and because of the sensitivity of the area to climate change, they are considered to be one of the most valuable high-resolution marine climate archives and have been successfully used to study climate variability in the tropics[3,11,16–18]. Varve thickness is about 1 mm or more during the YD–Holocene transition[18].

### Core and age model

Core MD03-2621 was retrieved during IMAGES cruise XI (PICASSO) aboard R/V Marion Dufresne in 2003 (Laj and Shipboard Party 2004). Cariaco cores have been collected under the regulations of the Ocean Drilling Program and the IMAGES coring programme. In this study, data from depths between 480 and 540 cm below the seafloor are presented, encompassing the YD–Holocene transition. A detailed age model for core MD03-2621 was established by Deplazes et al.[11] and is based on the cross-correlation of total reflectance to dated colour records from the Cariaco Basin[49,50]. For the studied interval, the original age model is based on a floating varve chronology anchored to tree ring data by matching [14]C data[49]. The age model for core MD03-2621 was further fine-tuned by correlation of reflectance data to the NGRIP ice core $\delta^{18}O$ record on the GICC05 age scale[11]. The transition from the YD to the Holocene is characterized by a decrease in the sedimentation rate from 1.4 to 0.5 mm year$^{-1}$.

To account for possible depth offsets during storage and subsampling, we matched sediment colour data expressed as greyscale (GS) to the reflectance data from Deplazes et al.[11] with the software QAnalySeries[51]. To enable comparison with our record, ages in Lea et al.[3] were corrected for the age difference between the sediment-colour-based midpoint of the YD–Holocene transition in their record (11.56 kyr b2k) and in data from Deplazes et al.[11] (11.673 kyr b2k). The start and end of the change in reflectance were determined by the RAMPFIT software[52].

### Sample preparation

Samples for MSI of molecular proxies were prepared as described in Alfken et al.[53]: the original core was subsampled by LL channels, from which X-ray pictures (Hewlett-Packard Faxitron 43855A X-ray cabinet) and high-resolution digital images (smart-CIS 1600 Line Scanner) were obtained. The LL channels were then cut into 5-cm pieces, which were subsequently freeze-dried, embedded in a gelatin:carboxymethyl cellulose (4%:1%) mixture and thin-sectioned on a Microm HM 505 E cryomicrotome. From each piece, one 60-μm-thick and one 100-μm-thick, longitudinal slice (spanning the whole 5 cm piece) were prepared and affixed to indium-tin-oxide-coated glass slides (Bruker Daltonik, Bremen, Germany) for MSI and elemental mapping, respectively. Slices for MSI were further amended with a fullerite matrix[54].

For all slices, a high-resolution picture was taken on a M4 Tornado micro-X-ray fluorescence spectroscopy system (Bruker Nano Analytics). This picture was used as a reference to set up elemental mapping and MSI analysis, and also for the 2D comparison of elemental and proxy data to sediment colour. Sediment colour is expressed as GS value. To account for differences between single slices, ΔGS was calculated as the difference between a value and the median GS of each individual slice. Very low GS values corresponding to areas devoid of sediment, identified by a black background, were excluded from analysis.

### Elemental mapping

Elemental mapping of 100-μm-thick slices was performed on a M4 Tornado micro-X-ray fluorescence spectroscopy system (Bruker Nano Analytics) equipped with a micro-focused Rh source (50 kV, 600 μA) with a polycapillary optic. Measurements were conducted under vacuum, with a resolution of 50 μm, two scans per spot and a scan time of 5 ms per scan. Data were initially processed and visualized with M4 Tornado Software version 1.3. XY matrices of relevant elements and sediment colour were imported into MATLAB (R2016b) for further processing. To assess the correspondence between sediment colour and elemental composition, for each 5-cm piece, signal intensities of Ca, Fe, Ti and Si in single spots were binned according to ΔGS and average intensities were calculated for each bin (Extended Data Fig. 5). The bin size was 5 GS units.

### Molecular proxy analysis by MSI

MSI was carried out on a 7T solariX XR Fourier transform ion cyclotron resonance mass spectrometer coupled to a matrix-assisted laser desorption/ionization source equipped with a Smartbeam II laser (Bruker Daltonik, Bremen, Germany). Analyses were performed in positive ionization mode selecting for a continuous accumulation of selective ions window of $m/z$ 554 ± 12. Spectra were acquired with 25% data reduction to limit data size. Spatial resolution was obtained by rastering the ionizing laser across the sample in a defined rectangular area at a 100-μm spot distance. Considering laminae thickness in the millimetre range[18], such raster resolution is suited for seasonally resolved SST reconstruction. Settings for laser power, frequency and number of shots were adjusted for optimal signal intensities before each measurement; typical values were 250 shots with 200 Hz frequency and 60% laser power. External mass calibration was performed in electrospray ionization mode with sodium trifluoroacetate (Sigma-Aldrich). Each spectrum was also calibrated after data acquisition by an internal lock mass calibration using the Na$^+$ adduct of pyropheophorbide $a$ ($m/z$ 557.2523), a chlorophyll $a$ derivative generally present in relatively young marine sediments. Around 20,000 individual spots were thereby obtained for every 5-cm slice, each spot containing information on the abundance of diunsaturated and triunsaturated $C_{37}$ alkenones needed to calculate the $U^{K'}_{37}$ SST proxy.

We provide a two-pronged approach to decode SST proxy information: (1) a downcore $U^{K'}_{37}$ profile is obtained by pooling alkenone data from coeval horizons, and results in SST reconstructions with annual resolution, and (2) 2D images of alkenone distribution are examined in conjunction with maps of sediment colour and elemental distribution to filter single-spot alkenone data for season of deposition.

### SST reconstruction with yearly resolution

For the downcore profile, MSI data were referenced to the X-ray image by the identification of three teaching points per 5-cm piece. Afterwards, the X-ray image was corrected for tilting of laminae in the LL channels. This was done by identification of single laminae in the X-ray image and selection of a minimum of four tie points per lamina. A detailed description can be found in Alfken et al.[9]. After applying the corresponding age model, downcore profiles were established with 1-year resolution: the intensity of the two alkenone species relevant to the $U^{K'}_{37}$ proxy ($C_{37:2}$ and $C_{37:3}$) were recorded for each individual laser spot and filtered for a signal-to-noise threshold of 3. Only spots in which both compounds were detected were further considered. Intensity values were then summed over the depth corresponding to 1 year. By pooling proxy data into 1-year horizons, the effect of changing sedimentation rate and, thereby, changing downcore resolution is minimized. If at least ten spots presenting both compounds were available for a single horizon, data quality criteria were satisfied[54] and a $U^{K'}_{37}$ value was calculated as defined by Prahl and Wakeham[22]:

$$U_{37}^{K'} = \frac{C_{37:2}}{C_{37:2} + C_{37:3}} \tag{1}$$

To apply the gas chromatography (GC)-based calibrations for the $U_{37}^{K'}$ proxy, MSI-based data were converted to GC equivalents. Therefore, after MSI, sediment slices were extracted for conventional proxy analysis. Sediment was scraped off the slide and extracted following a modified Bligh and Dyer procedure[55,56]. Extracts were evaporated under a stream of nitrogen, re-dissolved in *n*-hexane and analysed on a Thermo Finnigan Trace GC-FID equipped with a Restek Rxi-5ms capillary column (30 m × 0.25 mm ID). For each 5-cm piece, a ratio between the $U_{37}^{K'}$ values obtained by GC flame ionization detector analysis and MSI of the whole piece was calculated. The average ratio of all pieces for which GC-based values could be obtained was 1.194, with a standard deviation of 0.021.

$$U_{37\ GC-FID}^{K'} = 1.194 \times U_{37\ MSI}^{K'} \tag{2}$$

This ratio was used to calculate GC-equivalent $U_{37}^{K'}$ values, which were then translated into SST using the BAYSPLINE calibration[57]. The average standard error of the BAYSPLINE model is 0.049 $U_{37}^{K'}$ units (corresponding to 1.4 °C) for samples with SST below 23.4 °C, but increases at higher values (to up to 4.4 °C)[57]. This is explained by the fact that sensitivity of the $U_{37}^{K'}$ to SST (that is, the slope of the regression) declines at higher values. In the current dataset, the 95% confidence interval is, on average, ±3.6 °C. The analytical precision of MSI-based SST reconstructions for the $U_{37}^{K'}$, using at least ten data points, according to Alfken et al.[9], is about 0.3 °C. Sources of uncertainty are summarized in Extended Data Fig. 10a.

For frequency analysis, a continuous, annually spaced record was constructed by linearly interpolating 49 missing values. The record was subsequently detrended. Spectral analysis was performed with the REDFIT module[58] using a Hanning window (oversample 2, segments 2). Continuous wavelet transforms were applied to investigate changes in cyclicity over time, using the Morlet wavelet with code provided by Torrence and Compo[59] for MATLAB. All steps, except for the wavelet analysis, were performed with the PAST software[60].

For the assessment of the interannual variability, the SST record was band-pass-filtered for periods between 2 and 8 years. The record is based on 1-year binned data; seasonality is thereby nullified and the highest frequency to be evaluated (Nyquist frequency) corresponds to a period of 2 years. Variability of this time series was quantified by calculating the standard deviation of the band-pass-filtered $U_{37}^{K'}$ signal in 25-year intervals. To account for the potential impact of analytical precision on the observed signal (Methods, section titled 'The effect of changing sedimentation rate on reconstructed interannual SST variability during the YD–Holocene transition'), the variability experiment from Alfken et al.[9] was revisited. A sediment extract had been sprayed on an ITO slide and analysed by MSI. We then randomly selected *n* spots and obtained a $U_{37}^{K'}$ value for the summed intensities of these spots. Precision was calculated as the standard deviation of five replicate experiments for *n* = 1, 5, 10, 20, 30, 40, 50 and 60. Decreasing analytical variability with increasing number of observations was fitted to a curve ($R^2$ = 0.838) described by the equation

$$\text{Analytical variability} = 0.0741 \times \text{number of spots}^{-0.558} \tag{3}$$

On the basis of this equation, analytical variability for each horizon could be calculated on the basis of the number of values included (Extended Data Fig. 10b). The mean variability for each 25-year window was then subtracted from the observed variability in the band-pass-filtered signal and the resulting proxy values were translated to SST following the equation by Müller et al.[61]. Statistical significance of the change in corrected SST variability after 11.66 kyr b2k was assessed with a *t*-test.

## Assessment of SST seasonality

For the assessment of SST seasonality, alkenone intensities from individual spots were binned according to ΔGS, with a bin size of 1 unit. Spots were then separated into the categories upwelling season and non-upwelling season by identifying the threshold ΔGS value that maximized the difference between average SST in the bins above and below it. Furthermore, this value had to fulfill three conditions: (1) be higher (lighter) than the bins with the highest relative abundance of Ca, Ti and Fe, which is indicative of the dark sediments associated to non-upwelling season, (2) be lower (darker) than the bin with highest relative abundance for Si indicative of light sediment associated to the upwelling season and (3) the number of spots categorized as upwelling and non-upwelling had to account for at least 25% of total spots. If criteria 1 and 2 prevented criteria 3 from being fulfilled, a limit of 15% was set. After separating data into these two categories, data were processed separately as described above for the unfiltered dataset and a downcore temporal resolution of 5 years was applied. Seasonality was calculated as the difference between both records and thus represents the difference between 5-year average SST in the non-upwelling and upwelling seasons.

Shift in seasonality was fitted to two different ramps with the RAMP-FIT software[52]. An unconstrained approach and a constrained approach (in which the start and end points of the ramp were restricted to the intervals 11.725–11.8 kyr b2k and 11.6–11.675 kyr b2k) were applied. Negative values were excluded from this fitting. The resulting groups of data were compared by a Mann–Whitney rank test.

SST seasonality in the modern Cariaco Basin was calculated for the years 1980 to 2020 based on the HadISST dataset[62] by dividing monthly data from each year into two groups and searching for the largest difference between the average temperatures of both groups. Each group had to include at least three consecutive months. In 36 out of 41 years, the warm season was defined from May to November or from July to November.

## Decadal-scale to centennial-scale SST changes during the YD–Holocene transition and in the early Holocene

Annually reconstructed SST (average SST = 24.3 °C) remains relatively stable during the YD–Holocene transition. At around 11.4 kyr b2k, a warming trend is observed. Averaging all data before 11.39 kyr and after 11.37 kyr results in a warming from 23.9 ± 1.6 °C to 25.5 ± 1.4 °C. Trends identified by MSI are consistent with conventional $U_{37}^{K'}$ analyses performed in the present study and those previously reported by Herbert and Schuffert[23] on Ocean Drilling Program core 165-1002C (Extended Data Fig. 1). These authors observed a slight warming several hundred years after the transition into the Holocene, between about 11.53 and 11.32 kyr b2k.

Three prominent SST maxima are observed between about 11.50 and 11.45 kyr b2k. The average SST in these 50 years is 1.3 °C higher than in the 50 years before and after. These maxima are synchronous with the 11.4-ka cold event or PBO characterized by a negative excursion in $\delta^{18}O$ and reduced snow accumulation rates in Greenland ice cores[63] (Extended Data Fig. 2). The PBO coincides with the oldest of the Bond events, that is, pulses of ice rafting in the Northern Atlantic indicative of climatic deterioration[64].

A warm tropical response to the PBO would be supported by the lower-resolution foraminiferal SST record of Lea et al.[3], which shows two data points of increased SST shortly after the end of the YD–Holocene transition. To enable direct comparison, ages in Lea et al.[3] were corrected for the age difference between the sediment-colour-based YD termination midpoint in their record and in data from Deplazes et al.[11]. After this correction, these maxima correspond to 11.43 and 11.50 kyr b2k (Extended Data Fig. 2). Further, the SST maxima coincide with a short-lived change to lighter-coloured sediments. Hughen et al.[19] described a correlation between brief North Atlantic cold events, such

as the PBO, and changes in tropical primary productivity mediated by stronger upwelling that result in lighter sediments in the Cariaco Basin. Far-reaching effects of the PBO have previously been described in West Asia, with increased dust plumes being related to a southward shift of the westerlies[65].

The identification of the mechanisms behind a potential TNA response to the PBO is beyond the scope of this study. However, we wish to point out that high-resolution records are crucial to identify such events and to differentiate between underlying changes coinciding in time and, as in the present case, sharp signals that act on the same multidecadal timescales and can potentially be triggered by the same processes[66].

## The effect of changing sedimentation rate on reconstructed interannual SST variability during the YD–Holocene transition

Pooling proxy data into 1-year horizons establishes a constant sampling rate and thereby prevents potential effects of changing sedimentation rates. The onset of the Holocene in the Cariaco Basin sediments is characterized by a sharp decrease in sedimentation rates from 1.4 to 0.5 mm year$^{-1}$ (refs. [11,19]). Consequently, in the yearly pooled data, we observe a reduction in the number of values summed for each horizon (Extended Data Fig. 10b), as fewer laser spots fit into the thinner Holocene annual layers. At the same time, the mean intensity in each of these spots slightly increases, consistent with a relative increase of the contribution of haptophytes to primary production[20].

We have previously shown that the precision of MSI-based molecular proxy analysis is dependent on both the number of spots pooled per data point and the signal intensity in these spots[54]. All horizons used in the downcore record are above the established threshold of ten spots and proxy variability was shown to stabilize above this threshold[9,54]. However, as a decrease in the number of values per horizon might still result in lower analytical precision and contribute to higher signal variability, we corrected variability in the 2–8-year window with the estimated analytical variability (see equation (3)). With this correction, the magnitude of the described variability decreases across the record, but the trend towards higher interannual variability in the Holocene persists (Fig. 2c).

## Varve formation and alkenone deposition in the sediments of the Cariaco Basin during the YD–Holocene transition

Comparison of elemental maps and sediment colour (Extended Data Fig. 5) shows a consistent pattern of lamination across the YD–Holocene transition that results from the seasonal interplay of precipitation, upwelling and dominant phytoplankton community composition. Darker laminae represent the rainy, non-upwelling (summer/fall) season and are enriched in Fe and Ti from terrigenous material and Ca sourced from biogenic $CaCO_3$ produced by foraminifera or coccolithophores. Lighter laminae are characterized by high abundance of Si and correspond to the increased production of biogenic opal by diatoms during the upwelling (winter/spring) season[67]. This is in agreement with observations by Hughen et al.[18], who described the laminae couplets in the Cariaco Basin as representing annual cycles, whereby light laminae are an indicator of high productivity associated with the winter/spring upwelling season and dark laminae are an indicator of summer/fall runoff and accumulation of terrigenous material. Deplazes et al.[68] described a divergent origin of lamination for a deeper section of the YD, with light laminae being rich in calcareous and terrigenous elements characteristic for the summer season, whereas dark layers were enriched in Si and Br, indicative of diatoms and organic-walled primary producers characteristic for the more productive winter season. Such an alteration of the characteristic pattern of lamination is not observed in the late YD investigated here.

This blueprint of seasonality was used to assess the seasonal behaviour of alkenones. Alkenones were deposited throughout the year, as evidenced by the fact that the number of spots containing detectable amounts of both alkenone species are not restricted to the upwelling or non-upwelling seasons but distributed across a relatively wide range of GS values to both sides of the median (Extended Data Fig. 6). Average alkenone signal intensity is higher in the non-upwelling season, pointing to a preference of alkenone producers for this season and/or to a stronger dilution of the signal in the upwelling season. In regards to the $U^{K'}_{37}$ SST proxy distribution in light versus dark layers, our observations are in agreement with the ability to capture the seasonal SST cycle with alkenones in sinking particles in the modern Cariaco Basin[69].

## Effect of changing seasonality on YD and early Holocene SST records from the western TNA

Changing seasonality can contribute to explaining contrasting lower-resolution SST records in the western TNA during the YD and the early Holocene. The strong warming during the YD–Holocene transition recorded in the foraminiferal Mg/Ca record of the Cariaco Basin (Lea et al.[3]; Extended Data Fig. 1) might be reflecting the more robust thermohaline stratification and increasingly warmer non-upwelling seasons, given the preference of *Globigerinoides ruber* for this season.

*Globigerinoides ruber* (white), as used by Lea et al.[3], is considered to be a dominant species in the tropics, with a relatively uniform annual distribution. However, in the modern Cariaco Basin, upwelling leads to a distinct foraminiferal community composition and seasonal turnover[70], consistent with the notion of warm-water foraminifera narrowing their occurrence to the warmest season[71]. The relative abundance of *G. ruber* increases in the non-upwelling (warm) season but rarely exceeds 15%, whereas the upwelling season is clearly dominated by *Globigerina bulloides*[72,73]. *Globigerinoides ruber* fluxes are consistently lowest when upwelling is most vigorous, as expressed in annual minima in SST (Extended Data Fig. 9b). As upwelling during the YD and early Holocene was more intense than in the present[70], the preference of *G. ruber* for the summer (non-upwelling) season might have been even more pronounced.

The development of a stronger seasonality in the early Holocene would thus have led to a narrower temporal occurrence of *G. ruber* in the non-upwelling season, during which it would also be exposed to higher SST. The average SST difference between seasons obtained in our analysis can be converted into annual SST amplitude by assuming a sinusoidal curve. By doing so, we observe an increase in the seasonal amplitude of 1.5 to 1.9 °C (depending on the ramp fitted), which is similar to the warming described by Lea et al.[3].

This interpretation is in agreement with Bova et al.[46], who observed that most Holocene climate reconstructions are biased towards the boreal summer/fall and reflect the evolution of seasonal rather than annual temperatures. As discussed above, this is probably not true for the $U^{K'}_{37}$ index in the Cariaco Basin, as alkenones are deposited throughout the year. The suggested weakening of summer stratification during the YD (as compared with the Holocene) might, however, explain why the lower-resolution $U^{K'}_{37}$ records from the semi-enclosed Cariaco Basin show no or weaker warming[23] than other, open-ocean, tropical YD records[4], where the interplay of upwelling, freshwater input and stratification are less relevant to the SST signal.

## Data availability

Data are accessible in the Pangaea database under doi.pangaea.de/10.1594/PANGAEA.946440 and doi.pangaea.de/10.1594/PANGAEA.946442.

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

**Acknowledgements** We thank S. Alfken, T. Napier, I. Obreht and J. Pätzold for fruitful discussions and comments on an earlier draft and Z. Lu for sharing data on modelled ENSO variability. This research was supported by the European Research Council under the European Union's Horizon 2020 research and innovation programme, grant agreement no. 670115 ZOOMECULAR (K.-U.H.) and by Germany's Excellence Strategy (EXC-2077) project 390741603 'The Ocean Floor – Earth's Uncharted Interface'.

**Author contributions** Conceptualization: L.W., G.H.H., K.-U.H. Methodology: L.W., J.W. Software: L.W., B.B. Formal analysis: L.W. Investigation: L.W., J.W., B.B. Resources: G.H.H., K.-U.H. Writing, original draft: L.W., K.-U.H. Writing, review and editing: L.W., B.B., G.H.H., K.-U.H. Visualization: L.W. Funding acquisition: L.W., K.-U.H.

**Funding** Open access funding provided by Staats- und Universitätsbibliothek Bremen.

**Competing interests** The authors declare no competing interests.

**Additional information**
**Correspondence and requests for materials** should be addressed to Lars Wörmer.

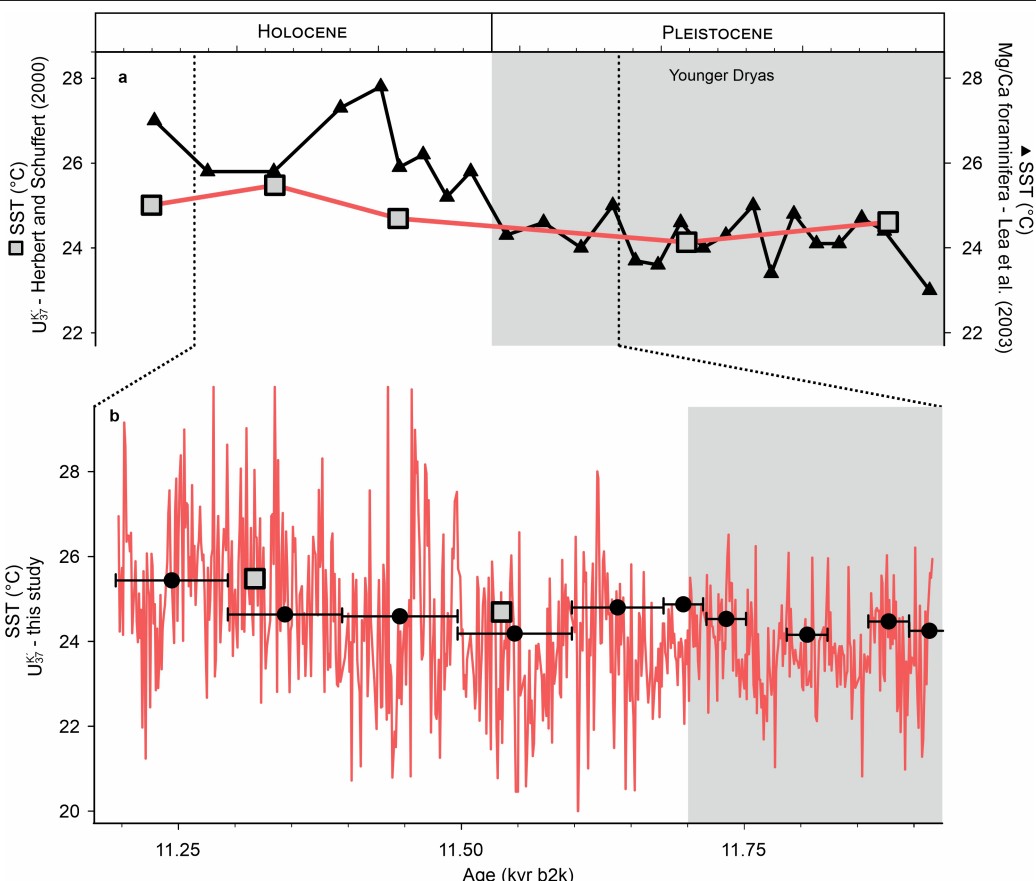

**Extended Data Fig. 1 | Comparison of MSI-based and conventional SST records in the Cariaco Basin. a**, Reconstructed SST from Herbert and Schuffert[23] based on the $U_{37}^{K'}$ proxy. The revised age model from Haug et al.[16] was applied and expressed in kyr b2k to make the record comparable with our data. Foraminifera-based SST reconstruction in the Cariaco Basin[3]. Ages were adjusted by normalizing to the respective midpoints of the YD–Holocene transition expressed as changes in sediment colour in Deplazes et al.[11] and Lea et al.[3]. **b**, Zoom-in to the YD–Holocene transition investigated in the present study, with MSI-based reconstruction (red line) shown together with data obtained by extracting sediment slices after having been used for MSI (black circles) and the two data points from the Herbert and Schuffert[23] study that fall into the investigated interval (grey squares). Horizontal bars indicate the age interval corresponding to the data pooled into each extraction.

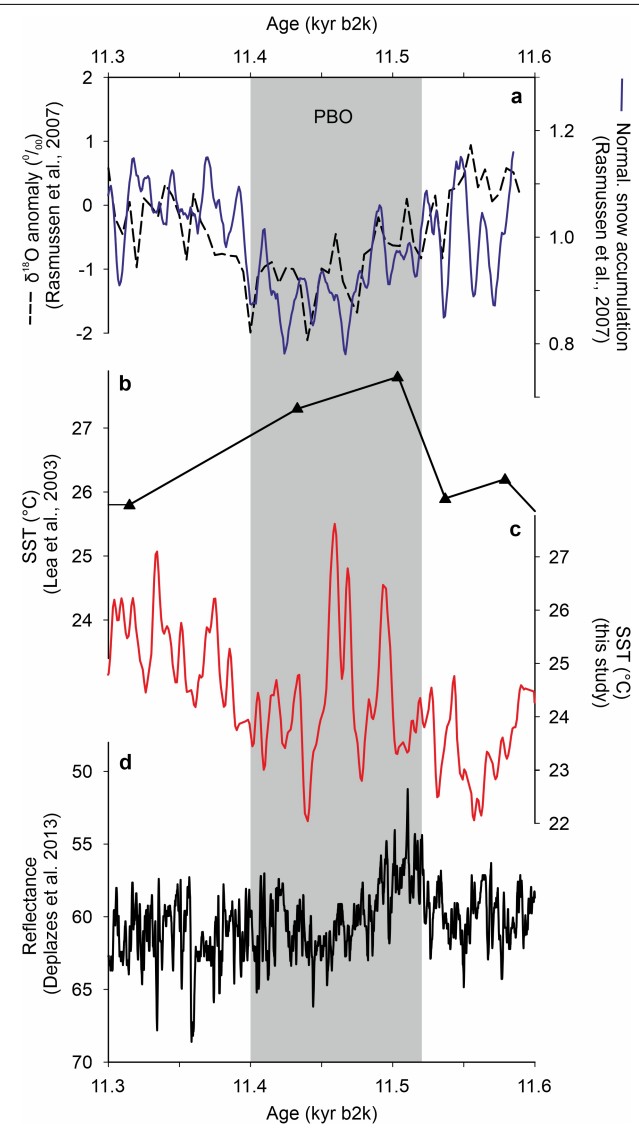

**Extended Data Fig. 2 | Evidence for a tropical response during the PBO.**
**a**, The PBO is expressed as a negative excursion in δ[18]O and reduced snow accumulation rates in Greenland ice cores[63]. **c**, Abrupt SST maxima in our record are consistent with lower-resolution SST reconstruction[3] (**b**) and are accompanied by an increase in reflectance indicative of changes in upwelling intensity[11] (**d**). Duration of the PBO (11.52–11.40 kyr b2k) as defined by Rasmussen et al.[13].

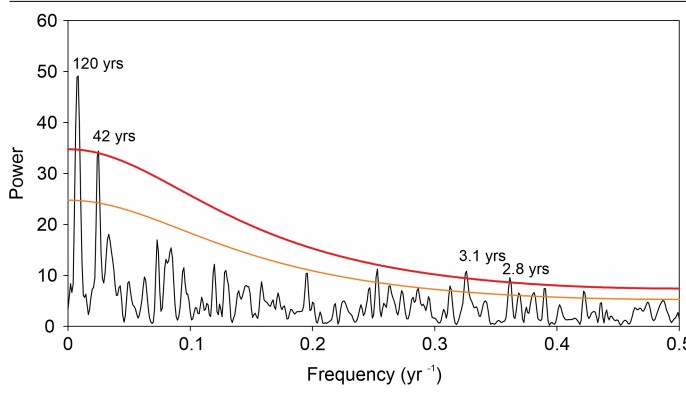

**Extended Data Fig. 3 | Spectral analysis of the annually resolved SST record.**
Thick red and thin orange lines are 99% and 95% significance levels against red
noise, respectively.

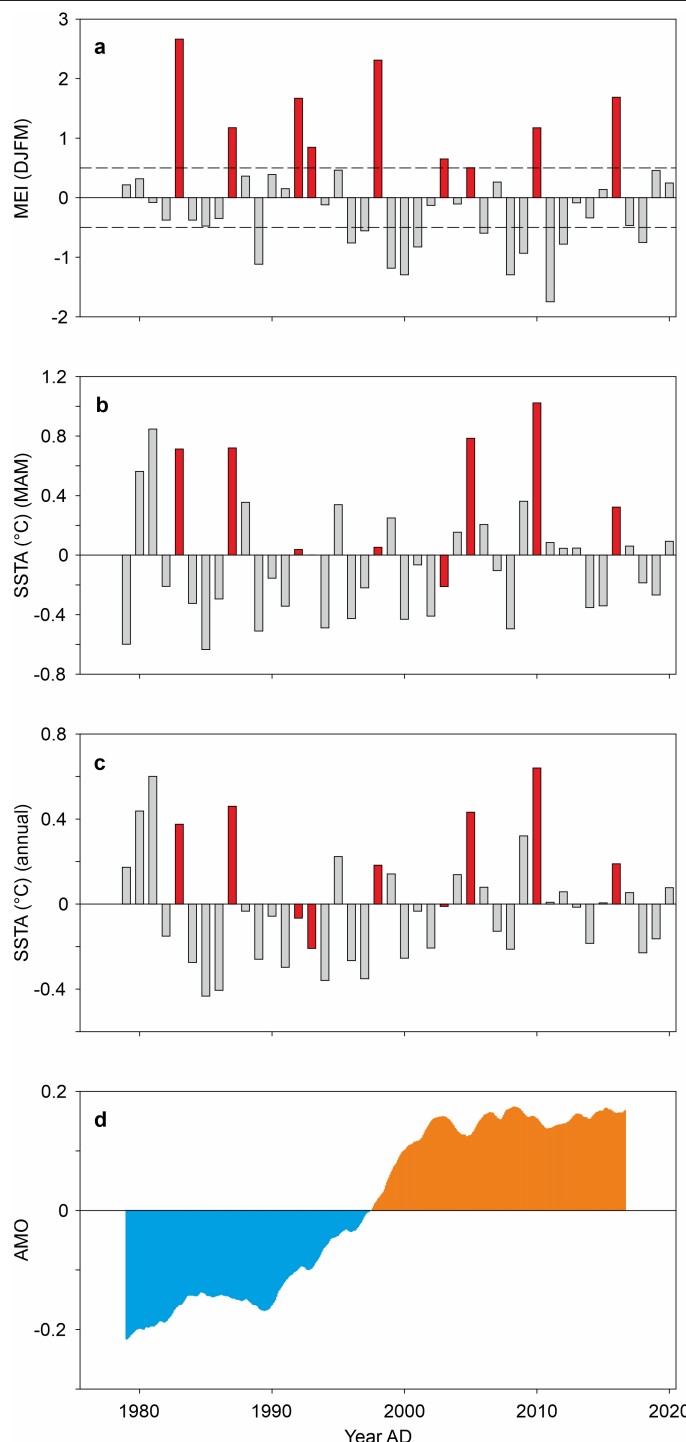

**Extended Data Fig. 4 | Teleconnection of ENSO to the Cariaco Basin.**
**a**, Positive ENSO events identified by the MEI.v2 index (https://psl.noaa.gov/
enso/mei/) are marked in red and lead to positive spring (**b**) and annual (**c**) SSTA
in the Cariaco Basin. SSTA are calculated from HadISST[62]. Four of the five (six)
strongest positive spring (annual) SST anomalies were related to positive ENSO
events. From the nine positive ENSO events detected, eight (six) resulted in
positive spring (annual) SST anomalies. **d**, The effect of positive ENSO was
muted between the years 1992 and 2003, coinciding with a shift from negative
to positive Atlantic Multidecadal Oscillation[74], which is consistent with TNA-
wide neutral response to positive ENSO events in 1992 and 2003 (ref. [75]) and
with a dependency of the ENSO teleconnection to the Atlantic Multidecadal
Oscillation state[76].

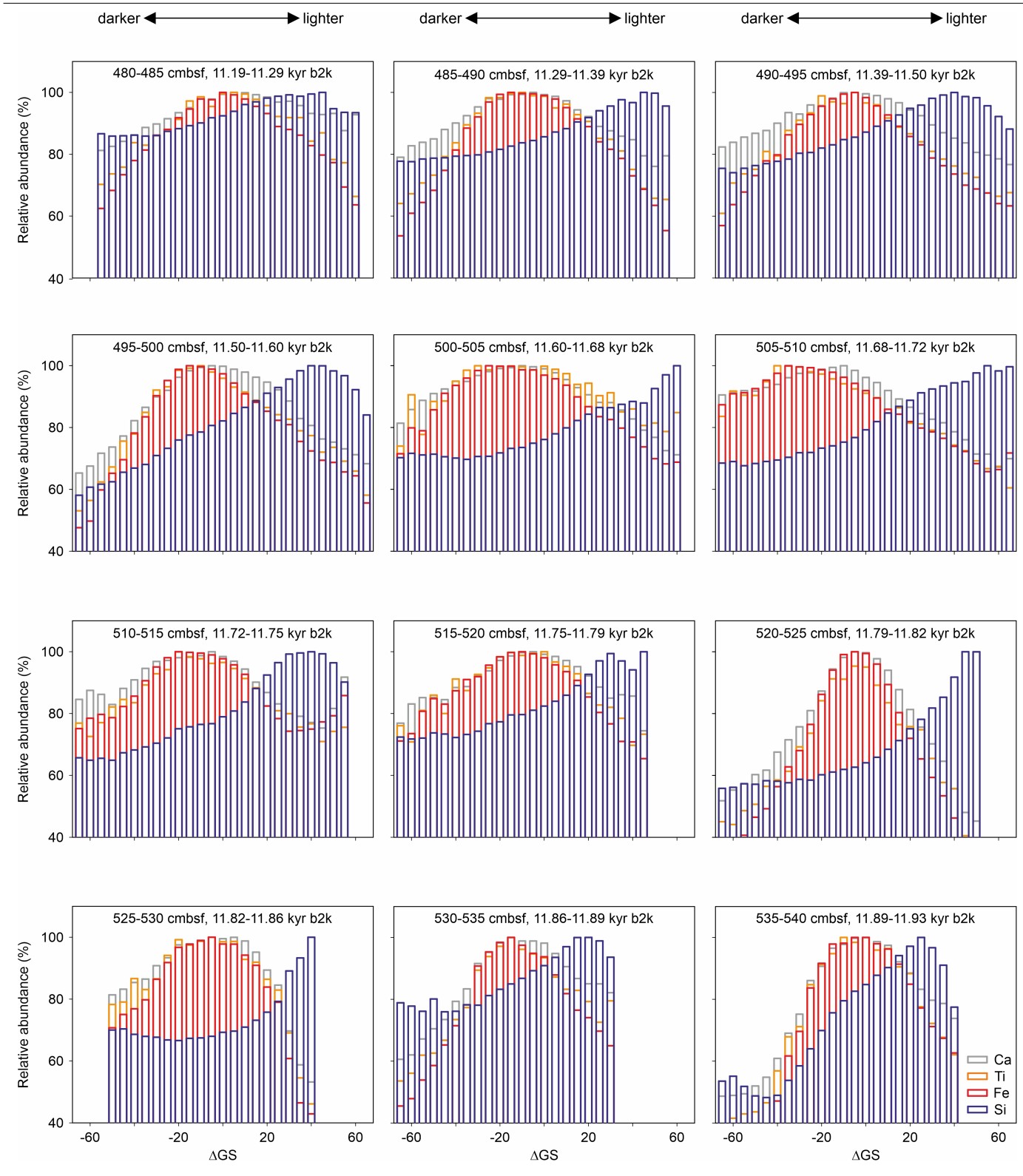

**Extended Data Fig. 5 | Elemental composition binned according to sediment colour.** Sediment colour is presented as delta greyscale (ΔGS, that is, the difference to the median GS value of each slice). Counts for each element were averaged for every bin and normalized to the highest value. Only bins with at least 100 data points were considered. The YD–Holocene transition, as defined by the change in sediment colour[11], is situated between 501 and 509 cmbsf (cm below seafloor).

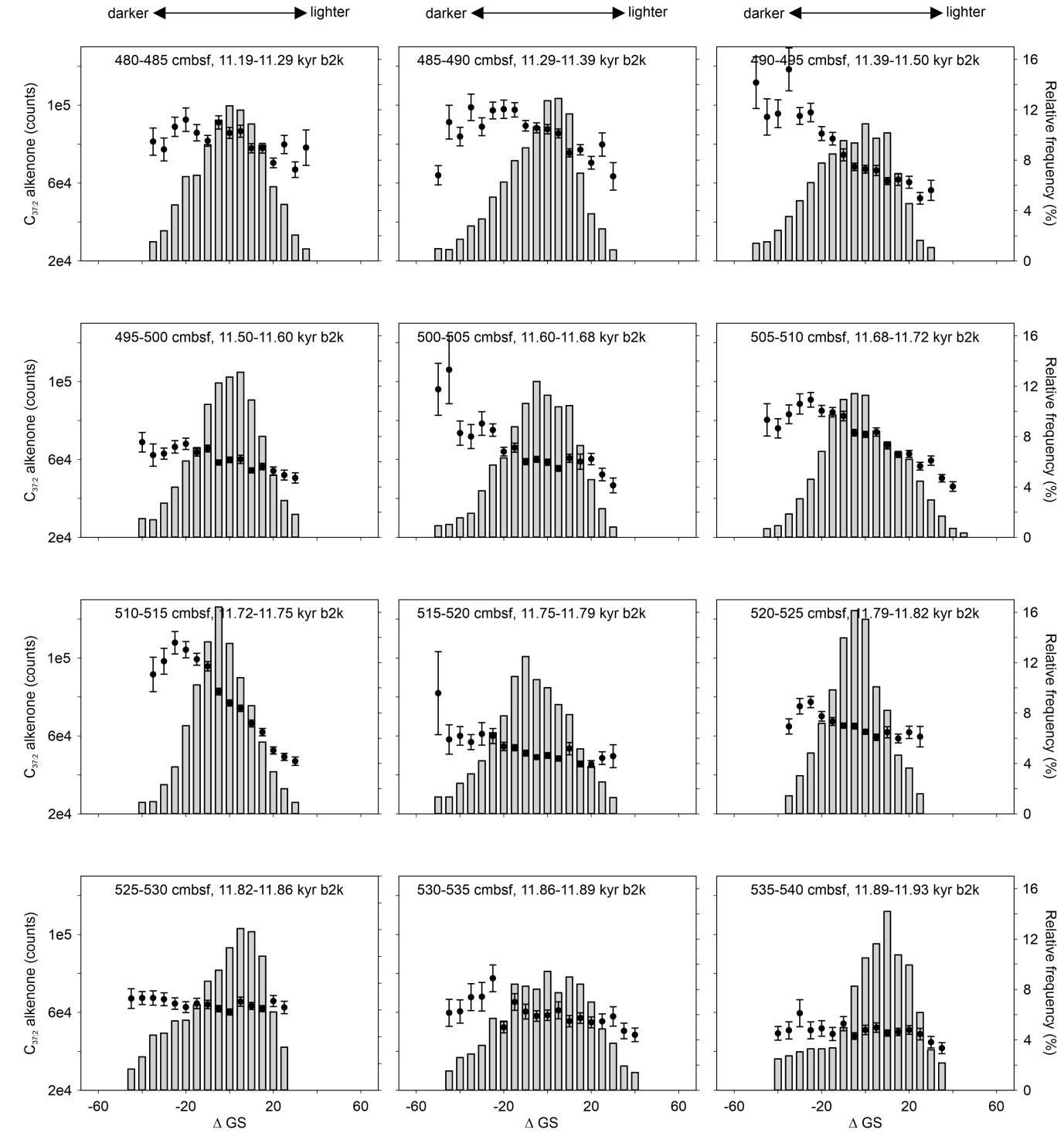

**Extended Data Fig. 6 | C$_{37:2}$ long-chain alkenone intensity binned according to sediment colour (black dots) and relative contribution of each bin to the total number of spots in which both alkenones used in the U$_{37}^{K'}$ proxy were detected (grey bars).** Sediment colour is presented as delta greyscale ($\Delta$GS, that is, the difference to the median GS value of each slice). The mean and standard error of alkenone intensity are shown. Only bins with at least 25 data points are shown. The YD–Holocene transition, as defined by the change in sediment colour[11], is situated between 501 and 509 cmbsf (cm below seafloor).

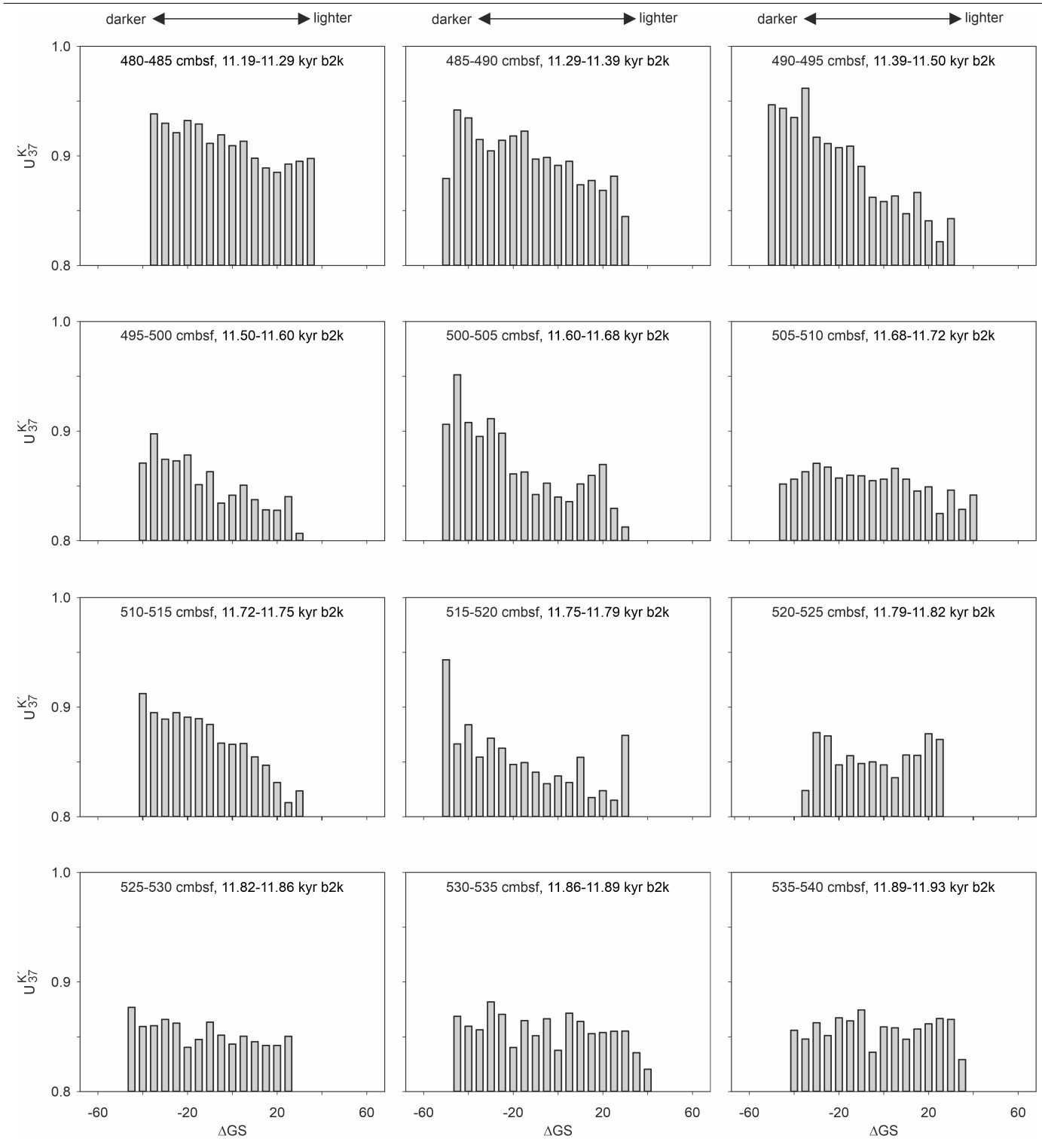

**Extended Data Fig. 7 | $U_{37}^{K'}$ values binned according to sediment colour.** Sediment colour is presented as delta greyscale ($\Delta$GS, that is, the difference to the median GS value of each sample). $U_{37}^{K'}$ was calculated on the basis of the sum of intensities in each bin and values obtained by MSI were converted to GC equivalents. Only bins with at least 25 data points are shown. The YD–Holocene transition, as defined by the change in sediment colour[11], is situated between 501 and 509 cmbsf (cm below seafloor).

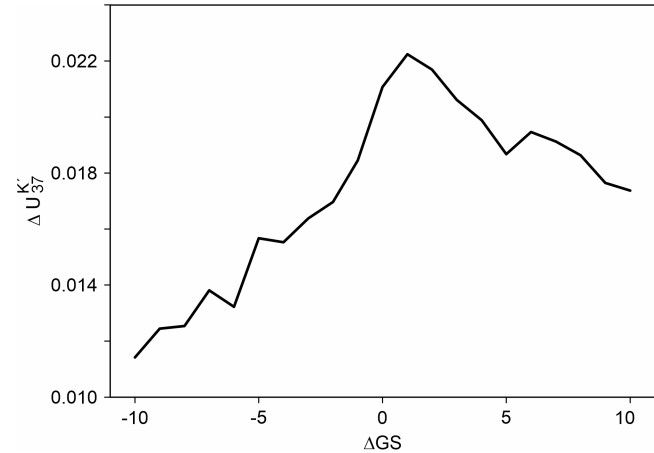

**Extended Data Fig. 8 | Impact of the sediment colour value used to separate $U^{K'}_{37}$ values into upwelling and non-upwelling seasons.** $\Delta U^{K'}_{37}$ denotes the increase in seasonality after 11.64 kyr b2k, which is a robust feature across a wide range of $\Delta$GS. $\Delta$GS refers to the difference in greyscale values to the threshold used in the main text. The increase in seasonality weakens as lower GS values are chosen, that is, more signals from the non-upwelling, warm layers are incorporated into the cold season. The strongest increase in seasonality is not detected at $\Delta$GS = 0 because we defined the threshold to be set in a way that at least 25% of the data points were included into each of the two seasons. Pushing the threshold towards higher values, in some cases assigning less than 25% of the data to the upwelling season, results in an even stronger enhancement of SST seasonality.

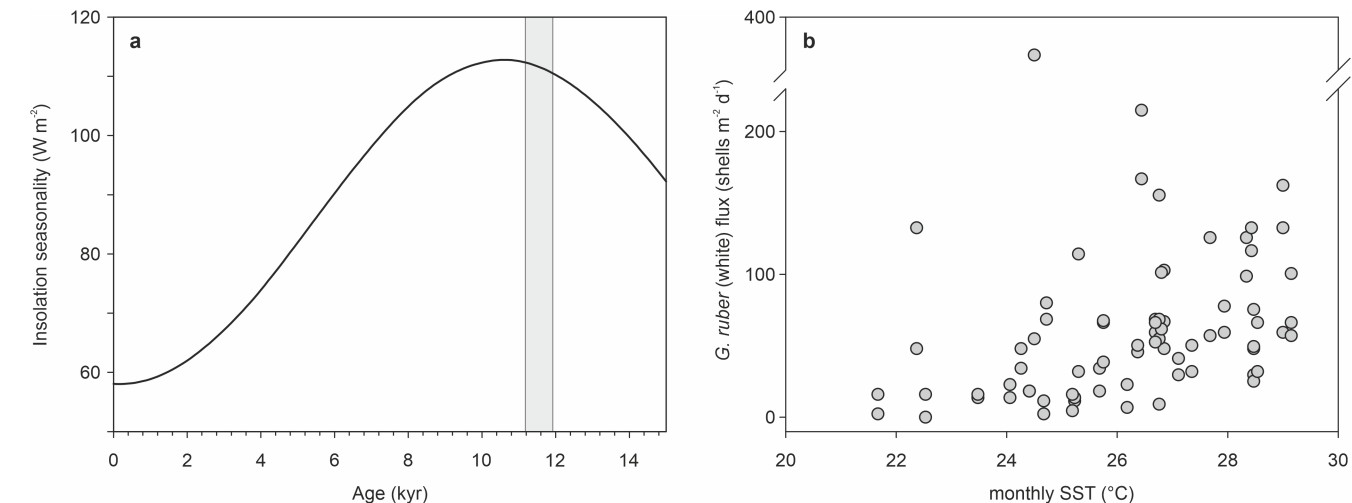

**Extended Data Fig. 9 | Seasonality of insolation and of foraminifera flux in the Cariaco Basin. a**, Seasonality of insolation at 10° N calculated as the difference between summer (JJA) and winter insolation (DJF) provided by Laskar et al.[77]. The shaded area indicates the investigated interval. **b**, Flux of *G. ruber* (white) in sediment traps from the Cariaco Basin plotted against monthly SST. Data from Tedesco and Thunell[73].

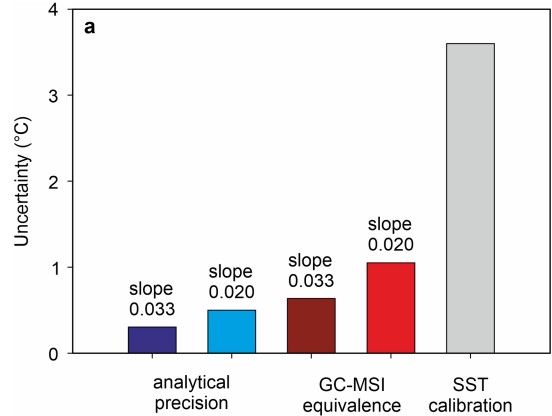

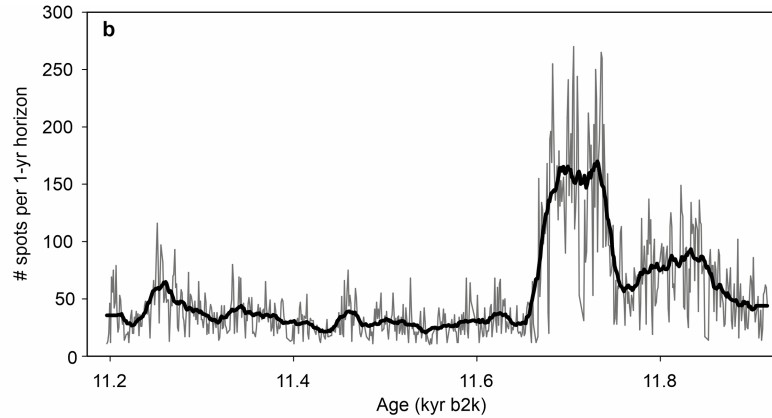

**Extended Data Fig. 10 | Assessment of error sources. a**, Sources of uncertainty in the reconstruction of SST based on MSI of the $U^{K'}_{37}$ proxy. Bars represent analytical precision based on the standard deviation of replicate measurements[9], the standard deviation of the average ratio between GC-based and MSI-based values in the current dataset, and the average 95% confidence interval for the SST calculation by means of the BAYSPLINE calibration model[57] in the current dataset. Uncertainties related to analytical precision and MSI to GC equivalence are expressed in °C using the slope (0.033) provided by Müller et al.[61] and using a 40% decrease in this slope to account for slope attenuation at high SST[57]. **b**, Number of spots pooled in each 1-year horizon to construct the $U^{K'}_{37}$ record. The thin grey line shows annual values and the thick black line shows the 25-year running mean.