## [Peer Review File · Nature]

Manuscript Title: Deglacial increase of seasonal temperature variability in the tropical ocean

Reviewer Comments & Author Rebuttals

Reviewer Reports on the Initial Version:

Referees' comments:

Referee #1 (Remarks to the Author):

The high resolution MSI-generated SST record from core MD03-2621 from the Cariaco Basin, and the new details it reveals about the Younger-Dryas to Holocene transition, are of immediate interest to several fields including oceanography, earth and atmospheric sciences, and marine geochemistry. The outstanding feature of the article is the application of the incredibly high resolution MSI technique to generate temperature records from lipid biomarkers which allows for original conclusions. This approach reveals otherwise invisible temperature information from the tropics during this time period and the technique deserves widespread attention. This data makes the paper a solid contribution, but the current version of the text needs careful polishing to highlight the exceptional aspects of this work to make it ready for publication. The current strength of this paper lies in the technique which allows for the discovery of previously hidden SST variability changes. The text should be modified to make it more clear 1) why is it remarkable that tropical SSTs were constant during the YD-Holocene transition; the abstract and requested summary specifically should reflect the points made in the main text to explicitly outline 2) why the discovery of seasonal amplitude changes is important, and 3) why understanding tropical decadal climate variability during this time is needed.

Specific comments

Line 18 of abstract refers to the “still controversial” imprint of the YD-Holocene transition on tropical temperatures as the first motivating factor of this work; the controversy is outlined in lines 50-52 (some records indicate YD was cooler (ref 3), while some indicate it was warmer (refs 4,17)). So I expected the first part of the paper and perhaps the first figure to give me a clear answer to the controversy. However, the main text does not directly address if (or how) the controversy is settled. It isn't until the last paragraph that the reader is pointed to section S4 – and it still not clear what the resolution is. Perhaps you are assuming the reader can see that “the average temperature was

not altered during the transition (line 22)” and “average reconstructed SST remains relatively stable during the YD-Holocene transition (Fig. 1A) (line 63)” IS the resolution (i.e., there wasn’t warming and there wasn’t cooling either) and so the controversy is settled? If that is the correct interpretation, perhaps restructure the paragraph starting at line 63 to make it clear that this goal was achieved. If that is not the correct interpretation, let the reader know what the resolution is and how the controversy was settled. Much attention is given to a post transition warming trend (lines 65-67), but isn’t that just to show the methodology is sound (and if so it would help to just say so)? Consider adding a note at the end of line 62 that the controversy is specifically addressed (rather than clumping everything into “provides insights”). Consider adding details (pre and post transition averages) to Fig. 1 to make it clear how this controversy was settled (is the answer warmer, cooler, or no change?). Finally, why is it important to settle this controversy and what do we gain? Some of these details are buried in the supplement, but it should be made obvious what the solution is and why it is important.

The next motivating factor of this work is the “poorly understood effects of the YD-Holocene transition on sub-annual to decadal climate variability”. These are more clearly addressed, Fig.2 tackles interannual variability and offers a mechanistic explanation (ENSO) plus the relevant text includes a significance statement about projecting future warming. Fig. 3 tackles seasonal variability and the relevant text offers a mechanism (ITCZ) and a significance explanation. For both the interannual and sub-annual aspects of this work – the proposed mechanism and relative significance should be highlighted in the abstract (and requested summary).

The main text would generally benefit from section headers/titles that highlight the main finding of each section to easily match with those listed in the abstract and figures. The better job you can do to hold the reader's hand and give them a good scaffold to follow, the more impactful this contribution will be.

Line 95 – Consider adding a helpful transition sentence, or descriptive phrases, to explain why this sentence reads almost exactly opposite of line 86 – I think you are trying to point out details in the timing, but without re-reading these paragraphs several times it seems like a typo in line 95 “YD ENSO amplitude was increased compared to Holocene” or line 86 “Holocene ENSO amplitude was increased compared to YD”

The ending is sudden and flat and would benefit from a summary paragraph (and perhaps an expansion of the current last paragraph).

Line 45-47. I realize this sentence follows the from the findings of the references noted in the sentences just prior, but it would be helpful to add citations in this sentence as well

Line 64: The phrase “and does not reflect the major environmental change” in the sentence is confusing. Currently the wording may indicate to several readers that there is NO environmental change at the transition, even though you are trying to simply describe the difference between the “boring SST” record and “striking precip” record. Changing the wording to something like: “Average reconstructed SST remained relatively stable during the YD-Holocene transition, unlike other major environmental changes that occurred , i.e., the northward shift of the ITCZ...” will help clarity here

Line 65-67: please provide (main text/fig or supplement txt/fig) some indication of this warming trend and/or indicate/clarify what data went into your averages (is 23.9 +/-1.6 before 11.39 ALL data obtained before 11.39? Is 25.5+/-1.4 all data after 11.37?) since it isn't obvious from the figures where the warming is and there are several hundred years of data before 11.39 compare to ~200 years of data after 11.37 from eyeballing Fig. 1. Even the very similar sentence in SI lines 17-19 is still not clear on what data in particular the averages are referring to, maybe specifically to the black dots in SI Fig. 1c closest to the Herbert squares?

Line 65-67: You repeat a lot of this text (and more) section S1, where does it belong? Certainly not both places word for word. SI Fig. 1 is remarkable to show the power of the MSI technique, does more of this, and possibly SI Fig. 1 belong in the main text?

Line 368 GC needs to be defined

Line 375 why not reference the figure for these measurements here? (and there are a few more places in the methods where reference to the extended data figs could be helpful)

Line 172 and Extended data Fig 5-7 – cmbssf needs to be defined somewhere, perhaps line 292 of main text or the figure legends themselves

Extended data Fig 5-7 – it would be helpful to add the age range under the cmbssf range to the top right corner of each subplot

Method section is solid with good detail (and more details would be welcome). Some of the info

about the study site seasonality might be nicely incorporated into the main text to make the current last paragraph of main text more fleshed out.

The reference to analytical precision (Line 378) is appreciated, but some discussion and/or representation of the errors on temperatures calculated from BAYSPLINE (due to errors in the calibration) is needed for Fig. 1A and Fig. 2C. Some estimation of errors in the approach to understanding seasonality (Fig. 3A) would be appreciated – there are several ways this can be tackled that involve the choices made to assign bins, colors, spots, etc. to test how sensitive the estimated YD and Holocene seasonality (black lines Fig. 3A) is to the process.

Referee #2 (Remarks to the Author):

This manuscript uses a new technique to study high resolution temperature changes in the varved Cariaco Basin, in order to detect seasonal to inter-annual variability in a 60 cm section of core (11.9-11.2ka), allowing the authors to capture new insights into climate variability across the deglacial, Younger Dryas and Holocene.

The method involved is mass spectrometry imaging (MSI) at 100- μ m resolution – citing reference 6, by the same author published in PNAS in 2014 – where they got GDGT data C_{Ca}T to generate temperatures for a different class of compounds those produced by archaea. Here the producers of alkenones are haptophytes, so this is a similar method but a different target analyte and producer – another exciting advance by this lab. They also cite the paper by Alfken (same research group) on the California margin, published in PP. They describe the methods used here, including the GCFID-MSI response slope which is close to 1:1 and statistical methods with care in the extended methods. They calibrate to temperature using the latest Bayesian calibration BAYSPLINE.

They also corroborate their values against conventional alkenone work (extended figure 1).

However, I find this comparison rather hard to look at, panel A and B have a lot more white space than is needed, the axis can zoom and lines in A and B can be overlain on one panel, so we can better read off the comparison of the classic methods and the warming found (minor). If possible the lines from AB, can also be plotted on the zoomed out view of C that shows the variability detected by this new methodology (red) and comparison extractions of sediment blocks (black symbols and bars) so the records can be compared on one plot

The novelty of the method and the exciting new insights merit publication in a high impact journal, as they will be of broad interest to those that follow the deglacial, methodological advances, climate

and questions that are often hampered by the time resolution of most archives.

The reviewer prompt asks about error bars, none are given for the y axis (temperature). for data that are based on quantification, these are typically minor and not replicates per sample but perhaps some overall measure of reproducibility for the different methods could be compared on the extended fig 1, where the T uncertainties differ between the compared proxies and methods, and calibrations. Uncertainties could also be represented in a bar graph for instrument, calibration etc.

The authors conclude that seasonal temperature changes are implicated in climate transitions, following ideas in Bova et al., and adding to these ideas with seasonal evidence from a very highly resolved archive.

The figures in the main text and supplement could almost all use improvement, they are functional figures, however the standard for visually appealing figures, e.g. sufficient line thickness (Fig 2 very thin lines) and color is not met, meaning that the data are hard to read, and for example the thin and thick red lines, of fig 1A, their main results are not shown to best advantage. A 5pt running mean is not appropriate, running means produce artefacts of smoothing by using a mean which is arectangular shape filter, the authors should be using something of a filter that doesn't produce artefacts like a hump shaped filter. Should fig. 1 be 1 page width so we can see the data at high resolution? Is there any reason why the figure should be vertically as tall (fig 1 and extended fig 1)? Fig 1. There is a lot of white space between records, Panel A is the important 1, panel C can be plotted using much less space perhaps overlain, also panel B. Try replotting for better use of space. EFig 1, panels A and B can be overlain etc.

Efig 2 – avoid the dots and dashes, there are many more plotting colors available to you. None of these data have any error bars?

Fig3/Efig5 – these washed out overlain bars, are very hard to read, improve aesthetics.

Referencing appears appropriate.

Abstract clarity could be improved, it passes the jargon expectations for paleoclimate for sure, but for a broader audience it may be a bit hard to follow and even for a paleoclimatologist, I see a leap between Cariaco and ENSO (a tropical Pacific AO phenomenon) and that leap is not explained. “We further observe modulations in interannual sea surface temperature variability that we attribute to a muting of the El Niño-Southern Oscillation at the end of the Younger Dryas, and a subsequent intensification during the early Holocene.”

Referee #3 (Remarks to the Author):

Summary:

This manuscript presents a sub-annually resolved sea surface temperature reconstruction from the Cariaco Basin based on the UK'37 proxy and mass spectroscopy imaging of the associated long-chain alkenones at 100-um resolution. The authors found that the seasonal amplitude of SSTs in the tropical North Pacific doubled during the transition from the Younger Dryas cold interval to the early Holocene (~12.9 to 11.7 ka), while mean annual SSTs remained essentially unchanged. They also found increased interannual variability across this transition, which they attributed to an intensification of ENSO variability. They claim that the results are consistent with model simulations demonstrating a forced climate response to meltwater and ice sheet forcing. They further claim that their results indicate that the termination of the Younger Dryas event occurred earlier in the tropical Pacific than the North Atlantic.

The manuscript presents a valuable set of well-dated SST reconstructions at unprecedented resolution from an established site that is likely to yield valuable insight into tropical Atlantic climate variability at the end of the last deglacial period. However, I have major concerns regarding several aspects of the data interpretation and conclusions, which are outlined below. On an organizational level, the writing also needs work in many places to improve clarity.

General comments:

One of the major issues I have concerns the interpretation of the change in interannual SST variability in the Cariaco Basin directly as a change in ENSO variability. While the teleconnection of ENSO to the tropical North Atlantic has indeed been well established in the literature, SST variability in the tropical North Atlantic also has clear interannual variations that are unrelated to dynamics in the tropical Pacific. In fact, Extended Data Fig. 4 (which the authors invoke to demonstrate the link between ENSO and SST anomalies in the Cariaco Basin) clearly shows this complexity. While the figure shows that four of the five strongest positive spring SST anomalies occurred during El Nino events, one of these events was associated with an El Nino event that barely met the threshold of a weak El Nino event, while several strong El Nino events (including the strong El Nino event of 1997-98), did not produce a significant temperature anomaly in the Cariaco Basin. In addition, only five of the nine El Nino events that occurred during the targeted 1980-2020 period produced substantial positive spring temperature anomalies in the Cariaco Basin (six out of nine for mean annual temperature anomalies). These data clearly show that ENSO teleconnections are only one of a myriad of factors that can produce SST anomalies in the Cariaco Basin on interannual timescales. For this reason, I strongly recommend that the authors reduce the degree of speculation in the

manuscript and reframe the interpretation simply in terms of a change in interannual SST variability in the tropical North Atlantic, which may have had links to tropical Pacific climate. Statements such as that in Line 115-117: “Our record provides a proxy-based, continuous evaluation of ENSO amplitude over the last major event of global warming and confirms its sensitivity to short term forcing” unnecessarily overreach and weaken the impact of their findings in my opinion. On another note of the SST reconstruction, the authors claim (e.g. Lines 82-85) that the data show a weakening of variability at sub-decadal frequencies during the last part of the Younger Dryas (followed by an increase during the late Holocene). While the increase in variance around 11.7 ka is clear in Fig. 2A-C, the decrease prior to that is far less obvious, and I do not think would pass a significance test given the noisy nature of low-frequency ENSO variability. The authors go on to interpret this decrease as reflecting a response to meltwater forcing at the end of the Younger Dryas, based on the TraCE model simulations. To justify such an interpretation, the authors need to first demonstrate whether the reduction in the proxy records is indeed significant.

Another major issue I have concerns the interpretation of the CCSM3 transient TraCE simulations. The authors claim in Line 95 that “Climate models have established that during most of the YD, ENSO amplitude was increased compared to the early Holocene, driven by the meltwater-induced collapse of overturning circulation.” The authors cite a single model simulation, which is the low-resolution (T31x3) transient CCSM3 simulation in Liu et al. (2014) (ocean model at nominal 3 degree resolution coupled to an atmosphere model at T31 resolution). Given the large biases in tropical Pacific climate in this low-resolution model and the widely varying representation of ENSO that exists across all climate models, a change in ENSO properties in any single model should be viewed as a model-dependent result until proven otherwise, especially in a low-resolution model such as this. Furthermore, the ENSO response in the TraCE simulation with the complete set of deglacial forcings does not agree with the UK’37 data (simulated ENSO variability decreases slightly following the termination of the Younger Dryas, in contrast to the increase in reconstructed variability around 11.7 ka). Of all the single forcing simulations that were performed as a part of the TraCE experiments, the authors plot the meltwater discharge simulation (red line in Fig. 2D) and the continental ice sheet forcing simulation (blue line in Fig. 2D) and claim that these simulations support the interpretation of a meltwater-driven decline in ENSO around 11.9 kya and an ice-sheet-driven increase in ENSO variability around 11.66 kya. I find these results to be unconvincing and based on selective interpretation of a subset of the available data.

In summary, I find the authors conclusions regarding ENSO to be poorly supported and largely speculative. As such, I find little evidence to support one of the authors’ main conclusions that their data provide support for a tropical Pacific trigger of the Younger Dryas termination (Lines 25-27 and 106-109). I think the manuscript would be dramatically improved by reframing the discussion to

focus on tropical North Atlantic SST variability, with a pared down section on possible dynamical links to the tropical Pacific. If the authors wish to incorporate the TraCE simulations into the discussion, these simulations need to be presented with more nuance regarding the areas of agreement and disagreement with the proxy data. A discussion on the limitations of the model should also be included.

A final comment I have is that given the novelty of the new results regarding the change in seasonality in SST in the tropical North Atlantic, the authors seem to miss the opportunity to draw a connection between the seasonal changes in SST and seasonal changes in the Atlantic ITCZ (as inferred from the sediment reflectance measurements of Deplazes et al., 2013). In several places (e.g. Line 146-149), the authors note that the reflectance data indicate changes in “the position of the ITCZ”. However, it is unclear whether the authors are interpreting the ITCZ changes as a northward shift in the mean annual position of the ITCZ, or as a shift of the seasonal range of the ITCZ in boreal summer/fall. Given that a large portion of the manuscript is dedicated to interpreting changes in seasonality in the SST reconstructions, it seems warranted to discuss how the inferred Atlantic ITCZ changes may also be interpreted in terms of seasonality. More generally, a sufficiently detailed discussion of the reflectance data in Fig. 1B is lacking (for instance, is there a large decrease in the variability of reflectance around 11.7ka, and if so, how do the authors interpret this change in light of their results?).

Author Rebuttals to Initial Comments:

Referees' comments:

Referee #1 (Remarks to the Author):

The high resolution MSI-generated SST record from core MD03-2621 from the Cariaco Basin, and the new details it reveals about the Younger-Dryas to Holocene transition, are of immediate interest to several fields including oceanography, earth and atmospheric sciences, and marine geochemistry. The outstanding feature of the article is the application of the incredibly high resolution MSI technique to generate temperature records from lipid biomarkers which allows for original conclusions. This approach reveals otherwise invisible temperature information from the tropics during this time period and the technique deserves widespread attention. This data makes the paper a solid contribution, but the current version of the text needs careful polishing to highlight the exceptional aspects of this work to make it ready for publication. **The current strength of this paper lies in the technique which allows for the discovery of previously hidden SST variability changes. The text should be modified to make it more clear 1) why is it remarkable that tropical SSTs were constant during the YD-Holocene transition; the abstract and requested summary specifically should reflect the points made in the main text to explicitly outline 2) why the discovery of seasonal amplitude changes is important, and 3) why understanding tropical decadal climate variability during this time is needed. *The reviewer's suggestions are excellent and we now explicitly state the environmental relevance of tropical high frequency SST variability and the projected increase of variability under global warming in the main text (l. 56-62) and the abstract (l. 19-22)***

We also more clearly point to the fact that average temperatures remained relatively stable as opposed to the increased high frequency variability and explain how changing seasonality might have impacted previous lower-resolution studies (l. 77-80, 172-175). We have added a summary paragraph in which we point out the climate information that has now become available, as well as the importance of these "previously hidden sources of SST variability"

Specific comments

Line 18 of abstract refers to the "still controversial" imprint of the YD-Holocene transition on tropical temperatures as the first motivating factor of this work; the controversy is outlined in lines 50-52 (some records indicate YD was cooler (ref 3), while some indicate it was warmer (refs 4,17)). **So I expected the first part of the paper and perhaps the first figure to give me a clear answer to the controversy.** However, the main text does not directly address if (or how) the controversy is settled. It isn't until the last paragraph that the reader is pointed to section S4 – and it still not clear what the resolution is. Perhaps you are assuming the reader can see that "the average temperature was not altered during the transition (line 22)" and "average reconstructed SST remains relatively stable during the YD-Holocene transition (Fig. 1A) (line 63)" IS the resolution (i.e., there wasn't

warming and there wasn't cooling either) and so the controversy is settled? If that is the correct interpretation, perhaps restructure the paragraph starting at line 63 to make it clear that this goal was achieved. If that is not the correct interpretation, let the reader know what the resolution is and how the controversy was settled. **Much attention is given to a post transition warming trend (lines 65-67), but isn't that just to show the methodology is sound (and if so it would help to just say so)?** Consider adding a note at the end of line 62 that the controversy is specifically addressed (rather than clumping everything into "provides insights"). **Consider adding details (pre and post transition averages) to Fig. 1 to make it clear how this controversy was settled (is the answer warmer, cooler, or no change?).** Finally, why is it important to settle this controversy and what do we gain? Some of these details are buried in the supplement, but it should be made obvious what the solution is and why it is important.

We agree that we should approach this controversy, and its resolution in a more straightforward way. Therefore, we now more explicitly state that average SST before and after the YD-Holocene remained unchanged, provide average values (l. 72-74), and "argue that the transition into the Holocene did not have an imprint on average SST, and that conflicting, low resolution SST records... can be explained by seasonal effects and changes to water column stratification" (l. 77-80). We then point to a more detailed discussion (l 172-175). We have also removed the distracting description of the post transition warming entirely to Supporting Information.

The next motivating factor of this work is the "poorly understood effects of the YD-Holocene transition on sub-annual to decadal climate variability". These are more clearly addressed, Fig.2 tackles interannual variability and offers a mechanistic explanation (ENSO) plus the relevant text includes a significance statement about projecting future warming. Fig. 3 tackles seasonal variability and the relevant text offers a mechanism (ITCZ) and a significance explanation. **For both the interannual and sub-annual aspects of this work – the proposed mechanism and relative significance should be highlighted in the abstract (and requested summary).**

We have added a sentence regarding the environmental relevance of high-resolution SST variability in the tropical Atlantic to the abstract (l. 19-22). However, based on the concerns of reviewer 3 and the suggestions of the editor, we have actually removed the ENSO-based, mechanistic explanation of increased interannual variability from the abstract. We provide an explanation of increased seasonality related to the position and seasonal range of the ITCZ which, in our opinion, is solid and has also not been questioned by the reviewers (l. 24-26). SST variability at both scales is stated in the new summary paragraph (l. 188-196)

The main text would generally benefit from section headers/titles that highlight the main finding of each section to easily match with those listed in the abstract and figures. The better job you can do to hold the reader's hand and give them a good scaffold to follow, the more impactful this contribution will be. **Section headers are now included**

Line 95 – Consider adding a helpful transition sentence, or descriptive phrases, to explain why this sentence reads almost exactly opposite of line 86 – I think you are trying to point out details in the timing, but without re-reading these paragraphs several times it seems like a typo in line 95 “YD ENSO amplitude was increased compared to Holocene” or line 86 “Holocene ENSO amplitude was increased compared to YD” ***Indeed, our intention was to point out that the general assumption of increased ENSO during the YD only is based on low resolution approaches, or single datapoints supposed to represent the whole episode, while we focus only on the last two centuries. We have tried to make this clearer now (l.105111).***

The ending is sudden and flat and would benefit from a summary paragraph (and perhaps an expansion of the current last paragraph).

Added

Line 45-47. I realize this sentence follows the from the findings of the references noted in the sentences just prior, but it would be helpful to add citations in this sentence as well

Added

Line 64: The phrase “and does not reflect the major environmental change” in the sentence is confusing. Currently the wording may indicate to several readers that there is NO environmental change at the transition, even though you are trying to simply describe the difference between the “boring SST” record and “striking precip” record. Changing the wording to something like: “Average reconstructed SST remained relatively stable during the YD-Holocene transition, unlike other major environmental changes that occurred, i.e., the northward shift of the ITCZ...” will help clarify here ***We agree and rephrased this sentence: “Despite the major environmental changes related to the northward shift of the ITCZ, which caused the abrupt change in sediment reflectance (Fig. 1b) and varve thickness^{11,19}, annually averaged SST remains constant across the YD-Holocene transition (Fig. 1a).”***

Line 65-67: please provide (main text/fig or supplement txt/fig) some indication of this warming trend and/or indicate/clarify what data went into your averages (is 23.9 +/-1.6 before 11.39 ALL data obtained before 11.39? Is 25.5 +/-1.4 all data after 11.37?) since it isn't obvious from the figures where the warming is and there are several hundred years of data before 11.39 compare to ~200 years of data after 11.37 from eyeballing Fig. 1. Even the very similar sentence in SI lines 17-19 is still not clear on what data in particular the averages are referring to, maybe specifically to the black dots in SI Fig. 1c closest to the Herbert squares?

Information has been added that all MSI-based data before/after 11.37 kyr b2k were included.

Line 65-67: You repeat a lot of this text (and more) section S1, where does it belong? Certainly not both places word for word. SI Fig. 1 is remarkable to show the power of the MSI technique, does more of this, and possibly SI Fig. 1 belong in the main text?

Based on an earlier comment, we have moved the discussion of the post-transition warming trend entirely to SI. We would prefer to not include SI Fig1 in the main text.

Line 368 GC needs to be defined

Done

Line 375 why not reference the figure for these measurements here? (and there are a few more places in the methods where reference to the extended data figs could be helpful)

We have added references to Ext Data Figs in lines 363, 419

Line 172 and Extended data Fig 5-7 – cmbsf needs to be defined somewhere, perhaps line 292 of main text or the figure legends themselves

As there is only one use of cmbsf in the methods and one in a figure legend, we have chosen not to use the abbreviation. In the Extended Data Figures, we define cmbsf in the figure legends

Extended data Fig 5-7 – it would be helpful to add the age range under the cmbsf range to the top right corner of each subplot

Good suggestion, age ranges have been added

Method section is solid with good detail (and more details would be welcome). Some of the info about the study site seasonality might be nicely incorporated into the main text to make the current last paragraph of main text more fleshed out.

We agree that shortly describing seasonality in the current Cariaco basin might help understand the suggested changes during the YD-Holocene transition, therefore as suggested by the reviewer, we have moved part of the site description to the main text (l. 155-161) We also have added an additional sentence explaining the potential bias of the study by Lea et al (2003) to the summer season, and, as suggested by the reviewer, a new, extended summary paragraph.

The reference to analytical precision (Line 378) is appreciated, but some discussion and/or representation of the errors on temperatures calculated from BAYSPLINE (due to errors in the calibration) is needed for Fig. 1A and Fig. 2C. Some estimation of errors in the approach to understanding seasonality (Fig. 3A) would be appreciated – there are several ways this can be tackled that involve the choices made to assign bins, colors, spots, etc. to test how sensitive the estimated YD and Holocene seasonality (black lines Fig. 3A) is to the process.

In response also to reviewer 2, we now state uncertainty of the BAYSPLINE model used for SST calculation in the method section (l. 413-417) and have added a new figure summarizing the three main sources of uncertainty: analytical precision, equivalence of GC and MSI-based data, and SST calibration (Extended Data Fig. 10a).

We also assess the impact of the selected sediment color threshold for the attribution of proxy data to upwelling/non upwelling seasons, and hence the calculation of seasonality (l. 142-144, Extended Data Fig 8).

Referee #2 (Remarks to the Author):

This manuscript uses a new technique to study high resolution temperature changes in the varved Cariaco Basin, in order to detect seasonal to inter-annual variability in a 60 cm section of core (11.9-11.2ka), allowing the authors to capture new insights into climate variability across the deglacial, Younger Dryas and Holocene.

The method involved is mass spectrometry imaging (MSI) at 100- μm resolution – citing reference 6, by the same author published in PNAS in 2014 – where they got GDGT data CCaT to generate temperatures for a different class of compounds those produced by archaea. Here the producers of alkenones are haptophytes, so this is a similar method but a different target analyte and producer – another exciting advance by this lab. They also cite the paper by Alfken (same research group) on the California margin, published in PP. They describe the methods used here, including the GCFID-MSI response slope which is close to 1:1 and statistical methods with care in the extended methods. They calibrate to temperature using the latest Bayesian calibration BAYSPLINE.

They also corroborate their values against conventional alkenone work (extended figure 1). However, I find this comparison rather hard to look at, panel A and B have a lot more white space than is needed, the axis can zoom and lines in A and B can be overlain on one panel, so we can better read off the comparison of the classic methods and the warming found (minor). If possible the lines from AB, can also be plotted on the zoomed out view of C that shows the variability detected by this new methodology (red) and comparison extractions of sediment blocks (black symbols and bars) so the records can be compared on one plot

We appreciate the reviewer's assessment of figure quality and suggestions to improve it. We have now modified Extended Data Fig.1 accordingly. Data from classic methods have been combined in panel A. We have zoomed-in on the y-axis, compressed panels and removed white space. Also, the size of the squares representing the Herbert and Schuffert data is now identical in panels A and B, which should allow the reader to easily compare the records. We have chosen not to overlay panels A and B in order to keep the larger zoom in panel B.

The novelty of the method and the exciting new insights merit publication in a high impact journal, as they will be of broad interest to those that follow the deglacial, methodological advances, climate and questions that are often hampered by the time resolution of most archives.

The reviewer prompt asks about error bars, none are given for the y axis (temperature). for data that are based on quantification, these are typically minor and not replicates per sample but perhaps some overall measure of reproducibility for the different methods could be compared on the extended fig 1, where the T uncertainties differ between the compared proxies and methods, and calibrations. Uncertainties could also be represented in a bar graph for instrument, calibration etc.

We now state uncertainty of the BAYSPLINE model used for SST calculation in the method section (l. 413-417) and have added a new figure summarizing the three main sources of uncertainty (Extended Data Fig. 11). In response to reviewer 1, we now also estimate the uncertainty associated to binning into upwelling/non upwelling seasons based on sediment color.

The authors conclude that seasonal temperature changes are implicated in climate transitions, following ideas in Bova et al., and adding to these ideas with seasonal evidence from a very highly resolved archive. The figures in the main text and supplement could almost all use improvement, they are functional figures, however the standard for visually appealing figures, e.g. sufficient line thickness (Fig 2 very thin lines) and color is not met, meaning that the data are hard to read, and for example the thin and thick red lines, of fig 1A, their main results are not shown to best advantage. A 5pt running mean is not appropriate, running means produce artefacts of smoothing by using a mean which is a rectangular shape filter, the authors should be using something of a filter that doesn't produce artefacts like a hump shaped filter. Should fig. 1 be 1 page width so we can see the data at high resolution? Is there any reason why the figure should be vertically as tall (fig 1 and extended fig 1)? Fig 1. There is a lot of white space between records, Panel A is the important 1, panel C can be plotted using much less space perhaps overlain, also panel B. Try replotting for better use of space.

Efig 1, panels A and B can be overlain etc.

Efig 2 – avoid the dots and dashes, there are many more plotting colors available to you. None of these data have any error bars?

Fig3/Efig5 – these washed out overlain bars, are very hard to read, improve aesthetics.

We have now tried to improve several of the figures in the main text and Extended Data. Figure 1 has thicker lines, and the smoothed line is now black for better visualization. Also, we no longer show a running average, but a Gaussian smoothing. We also combined panels B and C and reduced distance between panels to make it less tall.

Changes to Extended Data Fig 1 are described above. The dotted line in Ext Data Fig. 2 has been replaced by a solid blue line. Dashed line has been kept in order to also make both lines distinguishable if printed in greyscale. Thickness of lines has been increased. In Extended Data Figs 5-7 and Fig 3, line thickness has been increased for better visualization. We have tried several other approaches of plotting data in Fig 3 and Ext Data Fig 5 (filled bars, stacked bars, line plots), but in our opinion none of them improved the quality of the current one. All figures have been formatted according to the Guide for authors regarding, e.g., font size and panel labeling.

Referencing appears appropriate.

Abstract clarity could be improved, it passes the jargon expectations for paleoclimate for sure, but for a broader audience it may be a bit hard to follow and even for a paleoclimatologist, I see a leap between Cariaco and ENSO (a tropical Pacific AO phenomenon) and that leap is not explained. “We further observe modulations in interannual sea surface temperature variability that we attribute to a muting of the El Niño-Southern Oscillation at the end of the Younger Dryas, and a subsequent intensification during the early Holocene.”

We have rephrased the abstract. In agreement with reviewer 3, we have now removed the ENSO-based explanation from the abstract and discuss it in the main text in a more nuanced way. We have added information on the environmental relevance of high frequency SST variability that can be of interest to a broader audience

Referee #3 (Remarks to the Author):

Summary:

This manuscript presents a sub-annually resolved sea surface temperature reconstruction from the Cariaco Basin based on the UK'37 proxy and mass spectroscopy imaging of the associated long-chain alkenones at 100-um resolution. The authors found that the seasonal amplitude of SSTs in the tropical North Pacific doubled during the transition from the Younger Dryas cold interval to the early Holocene (~12.9 to 11.7 ka), while mean annual SSTs remained essentially unchanged. They also found increased interannual variability across this transition, which they attributed to an intensification of ENSO variability. They claim that the results are consistent with model simulations demonstrating a forced climate response to meltwater and ice sheet forcing. They further claim that their results indicate that the termination of the Younger Dryas event occurred earlier in the tropical Pacific than the North Atlantic.

The manuscript presents a valuable set of well-dated SST reconstructions at unprecedented resolution from an established site that is likely to yield valuable insight into tropical Atlantic climate variability at the end of the last deglacial period. However, **I have major concerns regarding several aspects of the data interpretation and conclusions, which are outlined below.** On an organizational level, the writing also needs work in many places to improve clarity.

General comments:

One of the major issues I have concerns the interpretation of **the change in interannual SST variability in the Cariaco Basin directly as a change in ENSO variability**. While the teleconnection of ENSO to the tropical North Atlantic has indeed been well established in the literature, **SST variability in the tropical North Atlantic also has clear interannual variations that are unrelated to dynamics in the tropical Pacific**. In fact, Extended Data Fig. 4 (which the authors invoke to demonstrate the link between ENSO and SST anomalies in the Cariaco Basin) clearly shows this complexity. While the figure shows that four of the five strongest positive spring SST anomalies occurred during El Niño events, one of these events was associated with an El Niño event that barely met the threshold of a weak El Niño event, while several strong El Niño events (including the strong El Niño event of 1997-98), did not produce a significant temperature anomaly in the Cariaco Basin. In addition, only five of the nine El Niño events that occurred during the targeted 1980-2020 period produced substantial positive spring temperature anomalies in the Cariaco Basin (six out of nine for mean annual temperature anomalies). These data clearly show that ENSO teleconnections are only one of a myriad of factors that can produce SST anomalies in the Cariaco Basin on interannual timescales. **For this reason, I strongly recommend that the authors reduce the degree of speculation in the manuscript and reframe the interpretation simply in terms of a change in interannual SST variability in the tropical North Atlantic, which may have had links to tropical Pacific climate**. Statements such as that in Line 115-117: “Our record provides a proxy-based, continuous evaluation of ENSO amplitude over the last major event of global warming and confirms its sensitivity to short term forcing” unnecessarily overreach and weaken the impact of their findings in my opinion.

We appreciate the reviewer’s constructive comments and agree that we may have overreached in our interpretation of SST variability being so strongly driven by ENSO. We now state the large variety of components shaping SST variability (l. 99-103), have removed or rephrased statements as the one pointed out by the reviewer, instead list ENSO as one factor that “might have contributed, amongst other factors, to reduced SST variability” (l.114-116), and thereby provide a more nuanced interpretation of the potential contribution of ENSO. We have also removed the link between interannual SST variability and ENSO from the abstract.

On another note of the SST reconstruction, the authors claim (e.g. Lines 82-85) that the data show a weakening of variability at sub-decadal frequencies during the last part of the Younger Dryas (followed by an increase during the late Holocene). While the increase in variance around 11.7 ka is clear in Fig. 2A-C, the decrease prior to that is far less obvious, and I do not think would pass a significance test given the noisy nature of low-frequency ENSO variability. The authors go on to interpret this decrease as reflecting a response to meltwater forcing at the end of the Younger Dryas, based on the TraCE model simulations. To justify such an interpretation, the authors need to first demonstrate whether the reduction in the proxy records is indeed significant.

We agree with the reviewer’s comment. The weakening of variability was based on the supposedly stronger ENSO during the YD argued for by proxy and model data, but cannot be

assessed based on our data. We therefore, and because of the other comments from the reviewer and the editor, have rephrased this section completely. We now just point out that variability was muted in the analyzed YD section compared to the Holocene. We then point out that ENSO in the very late YD could be different from the “canonical” YD response, which suggests a strengthened ENSO, and provide possible explanations why a relatively weak ENSO in the last centuries of the YD could be feasible (l. 105-116).

Arrows in figure 3 indicating weakening and subsequent strengthening of SST variability have been removed

Another major issue I have concerns the interpretation of the CCSM3 transient TraCE simulations. The authors claim in Line 95 that “Climate models have established that during most of the YD, ENSO amplitude was increased compared to the early Holocene, driven by the meltwater-induced collapse of overturning circulation.” The authors cite a single model simulation, which is the low-resolution (T31x3) transient CCSM3 simulation in Liu et al. (2014) (ocean model at nominal 3 degree resolution coupled to an atmosphere model at T31 resolution). Given the large biases in tropical Pacific climate in this low-resolution model and the widely varying representation of ENSO that exists across all climate models, a change in ENSO properties in any single model should be viewed as a model-dependent result until proven otherwise, especially in a low-resolution model such as this. Furthermore, the ENSO response in the TraCE simulation with the complete set of deglacial forcings does not agree with the UK’37 data (simulated ENSO variability decreases slightly following the termination of the Younger Dryas, in contrast to the increase in reconstructed variability around 11.7 ka). Of all the single forcing simulations that were performed as a part of the TraCE experiments, the authors plot the meltwater discharge simulation (red line in Fig. 2D) and the continental ice sheet forcing simulation (blue line in Fig. 2D) and claim that these simulations support the interpretation of a meltwater-driven decline in ENSO around 11.9 kya and an ice-sheet-driven increase in ENSO variability around 11.66 kya. I find these results to be unconvincing and based on selective interpretation of a subset of the available data.

Meltwater forcing of ENSO variability is now discussed from a conceptual standpoint, and two references for this hypothesis are provided, but we now refrain from plotting the TraCE model outcomes compared to our observations in figure 3 (panel D has been removed). We also point out that intensified ENSO during the central part of the YD is reflected in proxy data as well.

In summary, I find the authors conclusions regarding ENSO to be poorly supported and largely speculative. As such, I find little evidence to support one of the authors’ main conclusions that their data provide support for a tropical Pacific trigger of the Younger Dryas termination (Lines 25-27 and 106-109). I think the manuscript would be dramatically improved by reframing the discussion to focus on tropical North Atlantic SST variability, with a pared down section on possible dynamical links to the tropical Pacific. If the authors wish to incorporate the TraCE simulations into the discussion, these simulations need to be presented with more nuance regarding the areas of agreement and disagreement with the proxy data. A discussion on the limitations of the model should also be included.

We are grateful for the thorough and extremely useful review and hope to have arrived at a more balanced discussion of tropical North Atlantic SST variability, including a potential link to ENSO

A final comment I have is that given the novelty of the new results regarding the change in seasonality in SST in the tropical North Atlantic, the authors seem to miss the opportunity to draw a connection between the seasonal changes in SST and seasonal changes in the Atlantic ITCZ (as inferred from the sediment reflectance measurements of Deplazes et al., 2013). In several places (e.g. Line 146-149), the authors note that the reflectance data indicate changes in “the position of the ITCZ”. However, it is unclear whether the authors are interpreting the ITCZ changes as a northward shift in the mean annual position of the ITCZ, or as a shift of the seasonal range of the ITCZ in boreal summer/fall. Given that a large portion of the manuscript is dedicated to interpreting changes in seasonality in the SST reconstructions, it seems warranted to discuss how the inferred Atlantic ITCZ changes may also be interpreted in terms of seasonality. More generally, a sufficiently detailed discussion of the reflectance data in Fig. 1B is lacking (for instance, is there a large decrease in the variability of reflectance around 11.7ka, and if so, how do the authors interpret this change in light of their results?).

We now more clearly explain how ITCZ migration is coupled to the seasonal change of SST (l.149154), We also agree that the effects of a possible contraction of the annual ITCZ range was neglected. We now state that the observed environmental change recorded in the Cariaco Basin might originate in a change of the mean position of the ITCZ or a change to its annual range (e.g. abstract, l.24-26). We have refrained from further exploring reflectance data as these are not the main focus of the study, have been thoroughly interpreted in previous publications, and we fear that such an extension could make the manuscript less readable.

Reviewer Reports on the First Revision:

Referees' comments:

Referee #1 (Remarks to the Author):

Wormer et al. present extremely high resolution alkenone measurements to create a Northern Tropical Atlantic temperature record across the Younger Dryas-Holocene transition. They have modified the text to avoid speculating on the impacts of ENSO and focused the interpretation of the results [constant mean annual temperatures but increased seasonal variability (by comparing UK47 values to the seasonally impacted sediment grayscale values) across the transition] on the possible changes in ITCZ position or extent. This is important because changes in seasonal amplitudes can be rooted in large-scale changes in ocean-atmosphere forcing and/or more local circulation and productivity patterns, and as pointed out by Bova et al, seasonality can have large influence on proxy interpretation. After carefully reading the rebuttal, revised manuscript, methods, and supplement it is clear that this work has improved, and all reviewer comments have been addressed. Despite looking, I did not find any significant suggestions for improvement or change (only a few minor stylistic issues: reference #1 in the supplement is missing a year; line 414 in methods "SST below 23.4" should have a unit; line 77 you might consider adding the word 'annual' to "did not have an imprint on annual average SST"; line 33 the opening sentence would be more clear if the comma was replaced by "and"; and perhaps consider adding 'seasonal' to the title for "Deglacial increase of seasonal temperature variability in the tropical ocean" in order to distinguish annual and seasonal and give a nod to the impressively high resolution method). The text and figures are clear, uncertainties are addressed, and the introduction is excellent. The unique findings and nuanced interpretation are a valuable contribution.

Referee #2 (Remarks to the Author):

This is my second reading, I'm satisfied with the revisions and I recommend acceptance. I have three small suggestions:

Line 379, 20.000 should be written 20,000 or 20×10^3 .

Fig. 3b. something to improve the aesthetics and readability of the various data would be appreciated, try lines rather than bars.

Punchline - while the final sentence is a fine statement about what was wrong before, why not follow with a statement containing a more positive future outlook, about the potential scope for this MSI method as curious readers will want to know.

Can it be used for any cores or just varved cores? (I assume)

Can it be applied to 100m of core or just <1m as here? I assume it can be extended throughout, but why then just stop at 60cm here. What are the constraints - time, cost or sediment suitability?

Referee #3 (Remarks to the Author):

I commend the authors for thoroughly addressing my earlier comments, including reducing the degree of speculation regarding the discussion of interannual SST variability and its linkages to ENSO, as well as the reliance on the TraCE simulation. I also appreciated the expanded discussion of the seasonal SST data in association with the reflectance data and the possible role of the Atlantic ITCZ. I find the revised manuscript to be generally well-written and to contain important conclusions that are well-supported by the compelling results. Minor comments are listed below.

Line 52-53: It is stated that planktonic forams records a cooler YD. What type of proxy records record a "slightly warmer YD"? This information is needed to better understand the origins of the controversy regarding mean SST changes during the YD-Holocene transition. I also agree with Reviewer #1's initial comments that the authors do a good job pointing out this controversy, but then leave it unresolved until the next section. A sentence that foreshadows their conclusions, such as, "this discrepancy remains unresolved in low-resolution SST records".

Line 56: Replace "however" with "also", as both the mean annual SST changes (described in the preceding sentences), and the seasonal to interannual SST variability remain unresolved.

Line 70: Add the phrase "the high-resolution Uk37 reconstructions indicate" before "annual averaged SST remains constant across the YD-Holocene transition", so it's clear that this result came from this study.

Line 72: Replace "remains" with "remained". Here and elsewhere, be consistent with your verb tenses. You switch between the past and present tense in this sentence (and elsewhere) to describe your results.

Lines 81-85: You describe the coordinated timing between the three prominent SST maxima in the Uk37 data and the Preboreal Oscillation at 11.4 ka. More information is needed about this interpretation and significance of this result. This paragraph ends very abruptly as-is.

Line 83: Add "the" before "thermohaline circulation".

Line 90: Replace "associated to" with "associated with".

Line 91-92: The last sentence of the Fig. 1 caption is an incomplete sentence.

Line 96: Suggest replacing "Impact of" with "The variability at".

Line 127: Suggest replacing "accessed" with "assessed" or "reconstructed".

Line 145: Be consistent with your use of significant digits (0.79°C vs 1.8°C).

Line 184-185: I had to reread this sentence several times before I understood the meaning. To minimize confusion, I suggest a minor rewrite as follows: "Each bin encompasses 5 GS-units and

includes at least 25 successful Uk37' analyses".

Line 186: Suggest replacing "difference to the median" with "difference from the median".

Line 188-189: I find this sentence to be awkward. Consider revising to "... Last Interglacial thermal maxima are associated with large seasonal effects but weak annual SST changes". Also, are you describing global or tropical SSTs in this sentence?

Lines 189-193: I found the conclusions section to be rather lackluster. I suggest adding a few more details about your results (e.g. the directionality and magnitude) to deliver your key take-home points with more impact.

Author Rebuttals to First Revision:

Response to reviewers

Referee #1 (Remarks to the Author):

Wormer et al. present extremely high resolution alkenone measurements to create a Northern Tropical Atlantic temperature record across the Younger Dryas-Holocene transition. They have modified the text to avoid speculating on the impacts of ENSO and focused the interpretation of the results [constant mean annual temperatures but increased seasonal variability (by comparing UK47 values to the seasonally impacted sediment grayscale values) across the transition] on the possible changes in ITCZ position or extent. This is important because changes in seasonal amplitudes can be rooted in large-scale changes in ocean-atmosphere forcing and/or more local circulation and productivity patterns, and as pointed out by Bova et al, seasonality can have large influence on proxy interpretation. After carefully reading the rebuttal, revised manuscript, methods, and supplement it is clear that this work has improved, and all reviewer comments have been addressed.

Despite looking, I did not find any significant suggestions for improvement or change (only a few minor stylistic issues: reference #1 in the supplement is missing a year; line 414 in methods “SST below 23.4” should have a unit; line 77 you might consider adding the word ‘annual’ to “did not have an imprint on annual average SST”; line 33 the opening sentence would be more clear if the comma was replaced by “and”; and perhaps consider adding ‘seasonal’ to the title for “Deglacial increase of seasonal temperature variability in the tropical ocean” in order to distinguish annual and seasonal and give a nod to the impressively high resolution method).

The text and figures are clear, uncertainties are addressed, and the introduction is excellent. The unique findings and nuanced interpretation are a valuable contribution.

Thank you very much for your endorsement of our manuscript. We have implemented all of your suggestions. The addition of „seasonal“ to the title now leads to a total length of 76 characters, which we hope can still be accepted. Otherwise we would replace “increase” by “rise” in the title.

Referee #2 (Remarks to the Author):

This is my second reading, I'm satisfied with the revisions and I recommend acceptance.

Thank you very much for your endorsement of our manuscript.

I have three small suggestions:

Line 379, 20.000 should be written 20,000 or 20×10^3 .

Done

Fig. 3b. something to improve the aesthetics and readability of the various data would be appreciated, try lines rather than bars.

We have now removed information on Ti in the main figure as it is generally redundant with Fe; due to the reduction of overlaid bars, the readability should be improved.

Punchline - while the final sentence is a fine statement about what was wrong before, why not follow with a statement containing a more positive future outlook, about the potential scope for this MSI method as curious readers will want to know.

Can it be used for any cores or just varved cores? (I assume)

Can it be applied to 100m of core or just <1m as here? I assume it can be extended throughout, but why then just stop at 60cm here. What are the constraints - time, cost or sediment suitability?

We now state that assessing climate variability „ is now feasible via MSI-based analysis of molecular proxies and its combination with other high-resolution techniques“. Following the editor’s advice to keep any discussion at the end of the paper brief, we prefer not to include more methodological aspects regarding suitability of different sedimentary archives.

Referee #3 (Remarks to the Author):

I commend the authors for thoroughly addressed my earlier comments, including reducing the degree of speculation regarding the discussion of interannual SST variability and its linkages to ENSO, as well as the reliance on the TraCE simulation. I also appreciated the expanded discussion of the seasonal SST data in association with the reflectance data and the possible role of the Atlantic ITCZ. I find the revised manuscript to be generally well-written and to contain important conclusions that are well-supported by the compelling results.

Thank you very much for your endorsement of our manuscript and for your comments in the earlier round of revisions, which greatly contributed to improve the quality of the manuscript.

Minor comments are listed below.

Line 52-53: It is stated that planktonic forams records a cooler YD. What type of proxy records record a “slightly warmer YD”? This information is needed to better understand the origins of the controversy regarding mean SST changes during the YD-Holocene transition. I also agree with Reviewer #1’s initial comments that the authors do a good job pointing out this controversy, but then leave it unresolved until the next section. A sentence that foreshadows their conclusions, such as, “this discrepancy remains unresolved in low-resolution SST records”.

We now state that these records are “reconstructions based on molecular proxies and planktonic foraminifera”

Line 56: Replace “however” with “also”, as both the mean annual SST changes (described in the preceding sentences), and the seasonal to interannual SST variability remain unresolved.

The term however in this case was aiming at pointing to the fact that annual SST changes have been investigated (although with contradictory results), while higher resolution changes have not. However, we agree with the referee that both topics remain unresolved. Therefore we have now replaced “unresolved” with “unexplored”

Line 70: Add the phrase “the high-resolution Uk37 reconstructions indicate” before “annual averaged SST remains constant across the YD-Holocene transition”, so it’s clear that this result came from this study.

Done

Line 72: Replace “remains” with “remained”. Here and elsewhere, be consistent with your verb tenses. You switch between the past and present tense in this sentence (and elsewhere) to describe your results.

Corrected

Lines 81-85: You describe the coordinated timing between the three prominent SST maxima in the Uk37 data and the Preboreal Oscillation at 11.4 ka. More information is needed about this interpretation and significance of this result. This paragraph ends very abruptly as-is.

We have omitted the discussion of these maxima in the main text as it diverges from the main story we want to present. However, for interested readers, we offer a more detailed explanation as Supplementary Information. We now more explicitly direct the reader to this discussion: “These maxima are discussed in more detail in the Supplementary Information (section S1).”
“

Line 83: Add “the” before “thermohaline circulation”.

Done

Line 90: Replace “associated to” with “associated with”.

Done

Line 91-92: The last sentence of the Fig. 1 caption is an incomplete sentence.

Added “is indicated”

Line 96: Suggest replacing “Impact of” with “The variability at”.

Done

Line 127: Suggest replacing “accessed” with “assessed” or “reconstructed”.

Done

Line 145: Be consistent with your use of significant digits (0.79°C vs 1.8°C).

Done

Line 184-185: I had to reread this sentence several times before I understood the meaning. To minimize confusion, I suggest a minor rewrite as follows: "Each bin encompasses 5 GS-units and includes at least 25 successful Uk37' analyses".

Done

Line 186: Suggest replacing "difference to the median" with "difference from the median".

Done

Line 188-189: I find this sentence to be awkward. Consider revising to "... Last Interglacial thermal maxima are associated with large seasonal effects but weak annual SST changes". Also, are you describing global or tropical SSTs in this sentence?

Done

Lines 189-193: I found the conclusions section to be rather lackluster. I suggest adding a few more details about your results (e.g. the directionality and magnitude) to deliver your key take-home points with more impact.

We have now replaced „evolution of... SST variability“ by „strengthening of... SST variability“ and removed „and confirms its sensitivity to abrupt forcing“ in order to remove redundancy with the following sentence. Also, in response to referee 2, we have added an outlook on the importance of MSI. Following the editor’s advice to keep any discussion at the end of the paper brief, we prefer not to add any further details.